The paper has been revised to address the identified inaccuracies in the references.

# Distributional Pareto-Optimal Multi-Objective Reinforcement Learning

**Xin-Qiang Cai**[1]*, **Pushi Zhang**[2]*, **Li Zhao**[2], **Jiang Bian**[2],
**Masashi Sugiyama**[3,1]  **Ashley J. Llorens**[2]
[1] The University of Tokyo, Tokyo, Japan
[2] Microsoft Research Asia, Beijing, China
[3] RIKEN AIP, Tokyo, Japan

## Abstract

Multi-objective reinforcement learning (MORL) has been proposed to learn control policies over multiple competing objectives with each possible preference over returns. However, current MORL algorithms fail to account for distributional preferences over the multi-variate returns, which are particularly important in real-world scenarios such as autonomous driving. To address this issue, we extend the concept of Pareto-optimality in MORL into distributional Pareto-optimality, which captures the optimality of return distributions, rather than the expectations. Our proposed method, called Distributional Pareto-Optimal Multi-Objective Reinforcement Learning (DPMORL), is capable of learning distributional Pareto-optimal policies that balance multiple objectives while considering the return uncertainty. We evaluated our method on several benchmark problems and demonstrated its effectiveness in discovering distributional Pareto-optimal policies and satisfying diverse distributional preferences compared to existing MORL methods.

## 1  Introduction

Apart from most Reinforcement Learning (RL) works that consider the scalar reward of a task [1, 2, 3, 4], Multi-Objective Reinforcement Learning (MORL) has recently received extensive attention due to its adeptness at managing intricate decision-making issues with multiple conflicting objectives [5, 6, 7, 8]. In many multi-objective tasks, the relative preferences of users over different objectives are typically indeterminate a priori. Consequently, MORL's primary aim is to learn a variety of optimal policies under different preferences to approximate the Pareto frontier of optimal solutions. It has been demonstrated that MORL can significantly reduce the reliance on scalar reward design for objective combinations and dynamically adapt to the varying preferences of different users.

However, numerous real-world situations involve not only unknown relative preferences across multiple objectives, but also uncertain preferences in return distributions. These may include preference in risk [9, 10], safety conditions [11], and non-linear user's utility [12]. Consider, for instance, autonomous driving scenarios where agents need to strike a balance between safety and efficiency objectives. Different users might possess varying levels of risk tolerance. Some may demand high safety performance, tolerating lower expected efficiency, while others may seek a more balanced performance between safety and efficiency. Current MORL methods, by focusing exclusively on expected values with linear preferences, may not adequately capture multi-objective risk-sensitive preferences, hence being unable to deliver diverse policies catering to users with varied risk preferences.

In this work, we broaden the concept of Pareto-optimality in MORL to encompass distributional Pareto-optimality, which prioritizes the optimality of return distributions over mere expectations, as

---

*Equal contribution. Work done during an internship at Microsoft Research Asia.

37th Conference on Neural Information Processing Systems (NeurIPS 2023).

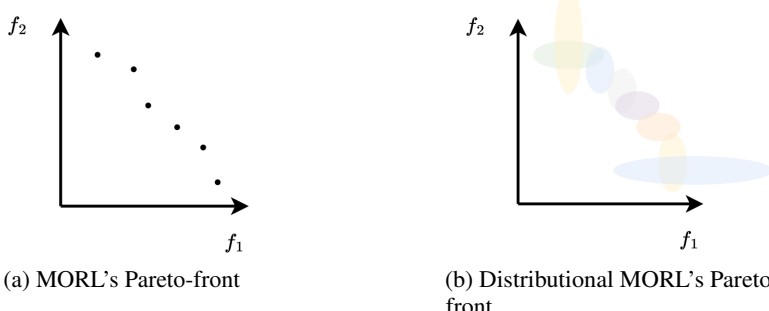

(a) MORL's Pareto-front  (b) Distributional MORL's Pareto-front

Figure 1: The comparison of learning targets between the traditional MORL tasks and distributional MORL tasks, in which $f_1$ and $f_2$ are two (conflicting) objectives.

depicted in Figure 1. From a theoretical perspective, we define Distributional Pareto-Optimal (DPO) policies, which capture the optimality of multivariate distributions through stochastic dominance [13, 14]. Distributional Pareto-Optimality delineates the set of policies with optimal return distributions, which we formally establish as an extension of Pareto-optimality in MORL.

On the practical side, we propose a novel method named *Distributional Pareto-Optimal Multi-Objective Reinforcement Learning* (DPMORL)[2], which aims to learn a set of DPO policies. We demonstrate that a policy achieving the highest expected utility for a given strictly increasing utility function is a DPO policy. To this end, we first learn diverse non-linear utility functions and then optimize policies under them [15]. Experimental outcomes on several benchmark problems attest to DPMORL's effectiveness in optimizing policies that meet preferences on multivariate return distributions. We posit that our proposed method significantly addresses the challenge of managing multiple conflicting objectives with unknown preferences on multivariate return distributions in complex decision-making situations.

Our main contributions are listed as follows:

1. We introduce the concept of Distributional Pareto-Optimal (DPO) policies. This expands the notion of Pareto-optimality in MORL to include preferences over the entire distribution of returns, not just their expected values. This enables agents to express more nuanced preferences over policies and align their decisions more closely with their actual objectives.

2. We propose DPMORL, a new algorithm for learning DPO policies under non-linear utility functions. This algorithm accommodates a broad range of distributional preferences, thus offering a more flexible and expressive approach to MORL.

3. We execute extensive experiments on various MORL tasks, demonstrating the effectiveness of our approach in learning DPO policies. Our findings show that our algorithm consistently surpasses existing MORL methods in terms of optimizing policies for multivariate expected and distributional preferences, underscoring its practical benefits.

## 2 Related Work

**Multi-Objective Reinforcement Learning.** MORL has emerged as a vibrant research area in the reinforcement learning community owing to its adeptness in managing intricate decision-making scenarios involving multiple conflicting objectives [5, 6]. A plethora of MORL algorithms have been put forth in the literature. For instance, [16] proposed the utilization of Generalized Policy Improvement (GPI) to infer a set of policies, employing Optimistic Linear Support (OLS) for dynamic reward weight exploration [17]. This GPI-based learning process was further enhanced by [7] through the incorporation of a world model to augment sample efficiency. [8] proposed an evolutionary approach to learn policies with diverse weights. However, our proposed approach diverges from these by leveraging utility functions to guide policy learning [18, 19]. The utility-based paradigm [6] accentuates the user's utility in decision-making problems, capitalizing on known information about the user's utility function and permissible policy types. Such methods scalarize the multi-dimensional objectives into a single reward function, enabling traditional RL algorithms to infer

---

[2]The code is available on `https://github.com/zpschang/DPMORL`.

desirable policies, while also respecting axiomatic principles where necessary. Several studies have utilized non-linear utility functions to guide policy learning [20, 12], catering to more intricate preferences compared to linear ones but still in view of the expectation of the Pareto front. In contrast, our work extends the Pareto-optimality to distributional Pareto-optimality by introducing a set of non-linear utility functions with distributional attributes to guide policy learning. Some of the recent works [21, 22, 23] in MORL defines Distributional Undominated Set or ESR set which is both very close to Distributionally Pareto-Optimal policies in this paper. These works propose novel designs to enhance policy improvement in distributional reinforcement learning algorithms to find the Distributional Undominated Set. Compared to these works, we build novel theoretical results on the conditions of DPO policies and a novel algorithm (DPMORL) of learning DPO policies based on reward shaping that can be built on the top of any online RL algorithm.

**Distributional Reinforcement Learning.** Distributional RL extends traditional RL by modeling the entire distribution of returns, rather than just their expected values [24, 25]. This approach has been shown to improve both learning efficiency and policy quality in a variety of single-objective RL tasks [26, 27]. Key algorithms in this area include Categorical DQN (C51) [24], Quantile Regression DQN (QR-DQN) [26], and Distributional MPO [27]. Despite the success of these distributional RL algorithms in single-objective settings, their direct extension to MORL has been limited due to the added complexity of handling multiple conflicting objectives and expressing preferences over the distribution of returns. On the other hand, the insights and techniques developed in distributional RL can provide a valuable foundation for incorporating distributional preferences into MORL, as demonstrated by our proposed approach.

**Risk-Sensitive and Safe Reinforcement Learning.** Risk-sensitive and safe RL are special cases of MORL that accentuate specific facets of reward distributions [6]. Risk-sensitive RL primarily considers reward variance, striving to optimize policies that negotiate the trade-off between expected return and risk [28, 29, 30]. Conversely, safe RL prioritizes constraints on the agent's behavior, ensuring adherence to specified safety criteria throughout the learning process [31, 32, 33]. Some studies have proposed the integration of constraints into the state space, constructing new Markov Decision Processes [11]. Others have explored the distributional aspects of rewards, investigating the implications of distributional RL on risk-sensitive and safety-oriented decision-making [26, 24]. However, our proposed setting is more general in terms of objectives, as it considers a broader range of user preferences and captures the entire distribution of rewards, rather than focusing solely on specific aspects such as risk, variance, or constraints. By extending MORL to incorporate distributional properties, our approach enables the learning of distributional Pareto-optimal policies that cater to diverse user preferences and offer better decision-making in a wide range of real-world applications.

## 3 Preliminaries: Multi-Objective Reinforcement Learning

Similar to the procedure of RL [34, 35], in MORL, the agent interacts with an environment modeled as a Multi-Objective Markov Decision Process (MOMDP) with multi-dimensional reward functions. A Multi-Objective MDP is defined by a tuple $(S, A, P, P_0, \boldsymbol{R}, \gamma, T)$, where $S$ is the state space, $A$ is the action space, $P$ is the state transition function, $P_0(s)$ is the initial state distribution, $\boldsymbol{R} : S \times A \times S \to \mathbb{R}^K$ is a vectored reward function and $K$ is the number of objectives, $\gamma$ is the discount factor, $T$ is the total timesteps[3]. The goal of the agent is to learn a set of Pareto-optimal policies, which represent the best possible trade-offs among the different objectives.

One popular methodology for MORL problems is the utility-based method, which combines the multi-dimensional reward functions into a single scalar reward function using a weighted sum or another aggregation method [36]. The intuition is to map the agent's preferences over different objectives to a scalar value for the agent training. Given a weight vector $\boldsymbol{w} = (w_1, w_2, \ldots, w_K)$, with $w_i$ representing the importance of the $i$-th objective, the scalarized reward function is defined as $r_{\text{scalar}}(s, a) = \sum_{i=1}^{K} w_i r_i(s, a)$. The agent then solves the scalarized MDP by optimizing its policy to maximize the expected scalar reward, using standard reinforcement learning algorithms like Q-learning or policy gradient methods. This approach can be straightforward to implement and has been shown to be effective in various MORL settings under expected preferences [37]. However, such a linear combination of each dimension of the reward cannot deal with the preferences with distributional considerations.

---

[3]We assume that current time $t$ is a part of state to accommodate for the finite horizon setting.

# 4 Distributionally Pareto-Optimal Multi-Objective Reinforcement Learning

In this section, we introduce our proposed theoretical framework and algorithms for extending MORL to handle distributional preferences.

## 4.1 Distributionally Pareto-Optimal Policies

In this work, we consider the reward as a $K$-dimensional vector, where each element represents the reward for a specific objective. Given the MOMDP and a policy $\pi$, we define the random variable of multi-objective return as $\boldsymbol{Z}(\pi) = \sum_{t=0}^{T} \gamma^t \boldsymbol{r}_t$, where $\boldsymbol{r}_t$ is the multi-dimensional reward at step $t$, and the states $s_0, s_1, \cdots, s_T$ and actions $a_0, a_1, \cdots, a_T$ are sampled from the MOMDP and the policy $\pi$ respectively. The return distribution of policy $\pi$, denoted as $\boldsymbol{\mu}(\pi)$, represents the joint distribution of the returns $\boldsymbol{Z}(\pi)$ following policy $\pi$. The utility function $f$ is a non-decreasing function that maps a $K$-dimensional return into a scalar utility value for the agent [38, 39, 40], capturing the user's distributional preferences over the different objectives. The expected utility of policy $\pi$ under the utility function $f$, represented as $\mathbb{E}_{\boldsymbol{z} \sim \boldsymbol{\mu}(\pi)} f(\boldsymbol{z})$, is the expected value of applying $f$ to the return distribution $\boldsymbol{\mu}(\pi)$.

Our goal is to learn a set of policies that are distributionally Pareto-optimal, which means that their return distributions of each policy cannot dominate that of another. To measure such a relationship, here we first introduce the concept of stochastic dominance:

**Definition 1** (Stochastic Dominance for Multivariate Distribution). *A multivariate distribution $\boldsymbol{\mu}_1$ dominates another distribution $\boldsymbol{\mu}_2$, denoted as $\boldsymbol{\mu}_1 \succ_{SD} \boldsymbol{\mu}_2$, if and only if $\boldsymbol{\mu}_1 \neq \boldsymbol{\mu}_2$ and for any non-decreasing utility function $f : \mathbb{R}^K \to \mathbb{R}$, $\boldsymbol{\mu}_1$ has greater expected utility than $\boldsymbol{\mu}_2$, i.e. $\mathbb{E}_{\boldsymbol{z} \sim \boldsymbol{\mu}_1} f(\boldsymbol{z}) \geq \mathbb{E}_{\boldsymbol{z} \sim \boldsymbol{\mu}_2} f(\boldsymbol{z})$.*

**Definition 2** (Stochastic Dominance for Policies). *A policy $\pi_1$ stochastically dominates another policy $\pi_2$, denoted as $\pi_1 \succ_{SD} \pi_2$, if and only if $\boldsymbol{\mu}(\pi_1) \succ_{SD} \boldsymbol{\mu}(\pi_2)$, indicating $\pi_1$ has greater expected utility than $\pi_2$ under any non-decreasing utility function $f$.*

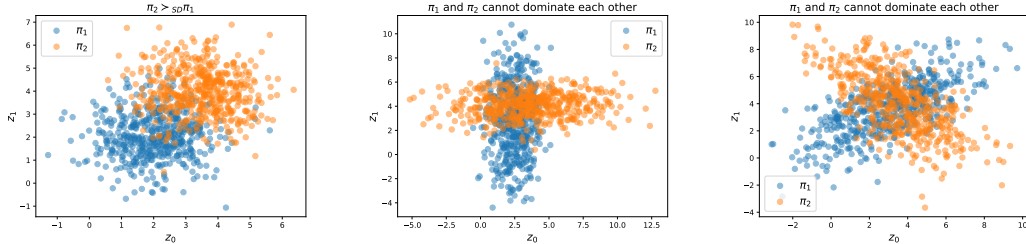

(a) Stochastic Dominance Example 1 (b) Stochastic Dominance Example 2 (c) Stochastic Dominance Example 3

Figure 2: Three 2D examples of stochastic dominance given synthetic returns of two policies. (a) $\pi_2 \succ_{SD} \pi_1$, since any non-decreasing utility function satisfies $\mathbb{E}f(\boldsymbol{Z}(\pi_2)) \geq \mathbb{E}f(\boldsymbol{Z}(\pi_1))$. (b) $\pi_1$ and $\pi_2$ cannot dominate each other. For example, there exists $f(\boldsymbol{z}) = \mathrm{ReLu}(z_0 - 5.0)$, such that $\mathbb{E}f(\boldsymbol{Z}(\pi_2)) > \mathbb{E}f(\boldsymbol{Z}(\pi_1))$, thus $\pi_1$ cannot dominate $\pi_2$. Similarly, $\pi_2$ cannot dominate $\pi_1$. Thus there is no dominant relationship between the two policies. (c) $\pi_1$ and $\pi_2$ cannot dominate each other, which is similar to (b).

The definition of stochastic dominance extends univariate first-order stochastic dominance [13] into multivariate cases. Figure 2 provides some examples of stochastic dominance given 2D returns of two policies. The definition of stochastic dominance in policies allows for comparing the optimality of distributions between policies, which allows us to define the Distributionally Pareto-optimal policy:

**Definition 3** (Distributionally Pareto-Optimal Policy). *Formally, $\pi_1$ is Distributionally Pareto-Optimal Policy if there does not exist a policy $\pi_2$ such that $\boldsymbol{\mu}(\pi_2) \succ_{SD} \boldsymbol{\mu}(\pi_1)$.*

In other words, $\pi_1$ is considered Distributionally Pareto-Optimal (DPO) if it is not stochastically dominated by any other policy. DPO policies are essential in our framework as they represent the most desirable policies for users with different distributional preferences. To find such a policy, we have the following theorem:

**Theorem 1.** *If a policy $\pi$ has optimal expected utility under some non-decreasing utility function $f$, where either utility function $f$ is strictly increasing, or $\pi$ is the only optimal policy with greatest expected utility under utility function $f$, then $\pi$ is a Distributionally Pareto-Optimal policy. Also, any Pareto-Optimal policy is a Distributionally Pareto-Optimal policy.*

The proof of Theorem 1 is provided in Appendix A which utilizes a form of optimal transport theory for stochastic dominance. In practice, this theorem guarantees that the optimal policy of a strictly increasing non-linear utility function is a DPO policy, making it a suitable candidate for deployment in MORL problems. The strict increasing property of utility function $f$ in Theorem 1 is crucial in the theoretical analysis for proving the distributional pareto-optimality. In the next section, we base on the result of this theorem to find optimal policies for a diverse set of utility functions, in order to learn the set of DPO policies. This result also shows that our definition of Distributional Pareto-Optimal policies is an extension of Pareto-optimal policies, allowing for a more diverse set of optimal policies to be captured.

We also formally prove that optimal policies for risk-sensitive and safe constraint objectives belong to the set of Distributional Pareto-Optimal policies. This proves that DPO policies can successfully cover policies with diverse distributional preferences. The detailed proof is provided by Theorem 3 in Appendix A.

## 4.2 Distributionally Pareto-Optimal Multi-Objective Reinforcement Learning

We now present our proposed algorithm, Distributionally Pareto-Optimal Multi-Objective Reinforcement Learning (DPMORL). The main idea of DPMORL is to learn a set of non-linear utility functions that can guide the agent to discover distributional Pareto-optimal policies. The algorithm proceeds in two stages: (1) generating the utility functions and (2) training the policies.

### 4.2.1 Utility Function Generation with Diversity-based Objective

The first component of our algorithm focuses on generating a diverse set of plausible utility functions, upon which we find the optimal policies to find a diverse set of optimal policies. This is essential to ensure our method can accommodate various distributional preferences and adapt to different problem settings. To achieve this, we propose (1) Non-decreasing Neural Network for parameterizing a diverse set of non-linear utility functions (2) an objective function of minimum distance which encourages generating a diverse set of utility functions.

**Non-decreasing Neural Network.** We employ non-decreasing neural network to parameterize the utility function. The use of a neural network allows us to represent complex, non-linear, and arbitrary continuous utility functions, while the non-decreasing constraint ensures that the utility function satisfies the desired properties for multi-objective problems. We ensure the non-decreasing property in neural networks by constraining the weight matrix to be non-negative and the activation function to be non-decreasing following existing work in convex neural networks [41] and QMIX [42], which can approximate any multivariate non-decreasing function with arbitrary small errors [43]. The implementation details of the Non-decreasing Neural Network are provided in Appendix B.

**Diversity-based Objective Function.** We propose an objective function based on diversity for learning a diverse set of plausible utility functions. Specifically, we define $f_{\theta_1}, f_{\theta_2}, \cdots, f_{\theta_M}$ to be the set of candidate utility functions, parameterized by $\theta_1, \cdots, \theta_M$. For any given utility function $f_{\theta_i}$, the learning objective is defined as

$$J^{\text{value}}(\theta_i) = \min_{j \neq i} \mathbb{E}_{\boldsymbol{z} \sim \mathcal{U}([0,1]^K)}[f_{\theta_i}(\boldsymbol{z}) - f_{\theta_j}(\boldsymbol{z})]^2 \tag{1}$$

$$J^{\text{grad}}(\theta_i) = \min_{j \neq i} \mathbb{E}_{\boldsymbol{z}_1, \boldsymbol{z}_2 \sim \mathcal{U}([0,1]^K)} \left[ \frac{f_{\theta_i}(\boldsymbol{z}_2) - f_{\theta_i}(\boldsymbol{z}_1)}{\|\boldsymbol{z}_2 - \boldsymbol{z}_1\|} - \frac{f_{\theta_j}(\boldsymbol{z}_2) - f_{\theta_j}(\boldsymbol{z}_1)}{\|\boldsymbol{z}_2 - \boldsymbol{z}_1\|} \right]^2 \tag{2}$$

$$J(\theta_i) = \alpha J^{\text{value}}(\theta_i) + (1 - \alpha) J^{\text{grad}}(\theta_i) \tag{3}$$

Optimizing this objective function can encourage diversity among the generated utility functions within the normalized range $[0, 1]^K$ in both value and derivative, leading to more comprehensive coverage of potential user preferences. We minimize the objective function by gradient descent on non-decreasing neural networks to generate a set of $N$ utility functions. To make the learned utility function strictly increasing for satisfying the constraints in Theorem 1, we add an additional term to the learned utility function as $\tilde{f}_{\theta_i}(\boldsymbol{z}) = f_{\theta_i}(\boldsymbol{z}) + \frac{\alpha}{K} \sum_{k=1}^{K} z_k$ in our experiment with $\alpha = 0.01$, and use $\tilde{f}_{\theta_i}(\boldsymbol{z}), i = 1, \cdots, M$ as the strictly increasing utility function to learn DPO policies.

#### 4.2.2 Optimizing Policy with Utility-based Reinforcement Learning

Once we have generated a diverse set of utility functions, the second component of our algorithm focuses on optimizing policies to maximize the expected utility. This process, which we call utility-based RL, leverages the generated utility functions to guide the optimization of policies. By focusing on the expected utility, our method can efficiently balance the trade-offs between multiple objectives and distributions, ultimately generating policies that are more likely to align with user preferences.

We show that the following utility-based reinforcement learning algorithm can effectively optimize the policy with respect to a given utility function.

---
**Algorithm 1** Utility-based Reinforcement Learning

---
**Input:** policy $\pi$, an environment $\mathcal{M} = (S, A, P, P_0, \boldsymbol{R}, \gamma, T)$, utility function $f$
**Output:** new policy $\pi'$
1: Augment state space with $\tilde{\mathcal{S}} = \mathcal{S} \times \mathcal{Z}$, where $\mathcal{Z}$ is the space of cumulative multivariate returns.
2: Let transition $\tilde{P}_0(\cdot)$ and $\tilde{P}(\cdot|(s_t, \boldsymbol{z}_t), a_t)$ with $s_0 \sim P(s_0), z_0 = 0$, and $s_{t+1} \sim P(\cdot|s_t, a_t)$, $\boldsymbol{z}_{t+1} = \boldsymbol{z}_t + \gamma^t \boldsymbol{r}_t$.
3: Let scalar reward function $R$ as $R((s_t, \boldsymbol{z}_t), a_t, (s_{t+1}, \boldsymbol{z}_{t+1})) = \gamma^{-t}[f(\boldsymbol{z}_{t+1}) - f(\boldsymbol{z}_t)]$.
4: Optimize policy $\pi$ under environment $\tilde{\mathcal{M}} = (\tilde{\mathcal{S}}, A, \tilde{P}, \tilde{P}_0, R, \gamma, T)$ under off-the-shelf RL algorithm (such as PPO or SAC) to $\pi'$.

---

Briefly, Algorithm 1 augments the state space with the cumulative multi-objective returns, and transforms the multi-dimensional rewards into a scalar reward by the difference in the utility function $f$. The following result shows that the new scalar-reward environment $\tilde{M}$ generated by Algorithm 1 has the same optimal policy as the optimal policy under utility function $f$:

**Theorem 2.** *The optimal policy $\pi^*$ under environment $\tilde{\mathcal{M}} = (\tilde{\mathcal{S}}, A, \tilde{P}, \tilde{P}_0, R, \gamma, T)$, with scalar reward function*

$$R((s_t, \boldsymbol{z}_t), a_t, (s_{t+1}, \boldsymbol{z}_{t+1})) = \gamma^{-t}[f(\boldsymbol{z}_{t+1}) - f(\boldsymbol{z}_t)]$$

*is the optimal policy under the utility function $f$, i.e.*

$$\mathbb{E}_{\boldsymbol{z} \sim \boldsymbol{\mu}(\pi^*)} [f(\boldsymbol{z})] = \max_{\pi} \mathbb{E}_{\boldsymbol{z} \sim \boldsymbol{\mu}(\pi)} [f(\boldsymbol{z})].$$

An advantage of Algorithm 1 is that we can directly utilize off-the-shelf RL algorithms, such as PPO [44] and SAC [45] without any modification, which makes the algorithm easy to implement using widespread existing implementations of online RL algorithms.

It is also important to note that our algorithm simplifies to optimizing the weighted sum of rewards in MORL when the utility function is linear. This implies that our method is a generalization of the linear utility function MORL approaches by accommodating a wide range of non-linear utility functions. This flexibility makes our algorithm particularly suited for problems where the user's preferences may not be adequately captured by a linear utility function.

In the next experimental section, DPMORL only undergoes a single iteration: we initially generate a set of $N$ utility functions as per the methodology detailed in Section 4.2.1. Subsequently, it optimizes a set of $N$ policies using these generated utility functions, as outlined in Section 4.2.2.

## 5 Experiments

In this section, we conducted several experiments under the setting of MORL. Through the experiments, we want to investigate the following questions:

1. Can Utility-based RL effectively learn policies with diverse distributional preferences?

2. Can DPMORL generate a set of diverse non-linear utility functions?

3. Can DPMORL obtain promising performance compared with state-of-the-art MORL methods in view of expected and distributional preferences?

## 5.1 Case Study of Utility Functions

To answer the first question, we train policies with a diverse set of utility functions on DiverseGoal, DeepSeaTreasure and Reacher. Here, we focus on showing the effectiveness of the Utility-based RL algorithm (Algorithm 1). We select several different non-linear functions $f$, each in favor of distributions at one target. We train policy $\pi_i$ with utility function $f_i$ with Algorithm 1 for each target, and show the policy's trajectory and return samples.

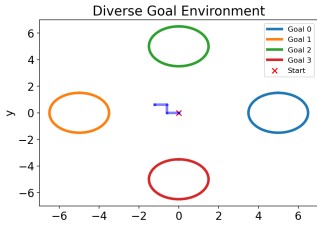

(a) The map of the DiverseGoal environment.

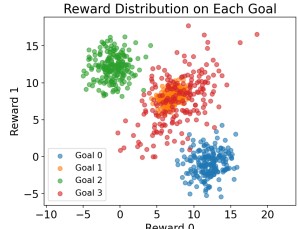

(b) Reward distribution within each goal position.

Figure 3: Illustrations of the DiverseGoal environment.

DiverseGoal is a MORL environment in Figure 3 with multiple goals that the agent can take multiple steps to reach, where each goal has its unique reward function. Upon reaching a particular goal, the agent secures a 2D reward. This reward is sampled from the specific normal distribution associated with that goal, as illustrated in Figure 3b. Conversely, if the agent reaches the boundary of the map, it incurs a negative 2-dimensional reward. The environment requires the agent to navigate the trade-offs between various reward distributions, underscoring the complexity and nuance inherent to the task.

In this study, we investigate if the learned policy can reach the goal that yields the highest expected utility under the utility function. The results are illustrated in Figure 4. Under different types of utility functions and return distributions, the utility-based RL algorithm is able to find the optimal policy with the highest expected utility, which shows the effectiveness of the utility-based RL algorithm.

DeepSeaTreasure and Reacher are among the most widely used MORL environments. Under the DeepSeaTreasure environment and Reacher environment, we show the return distributions of the policies optimized with utility-based RL under different utility functions. In Figure 5 and Figure 6, we use utility functions generated by DPMORL in Section 4.2.1 and a set of manually constructed utility functions for different types of distributional preferences respectively. From the result in Figure 5 and Figure 6, we can see that our algorithm effectively finds policies that maximize the expected utility for different non-linear utility functions. Meanwhile, the DeepSeaTreasure environment aligns well

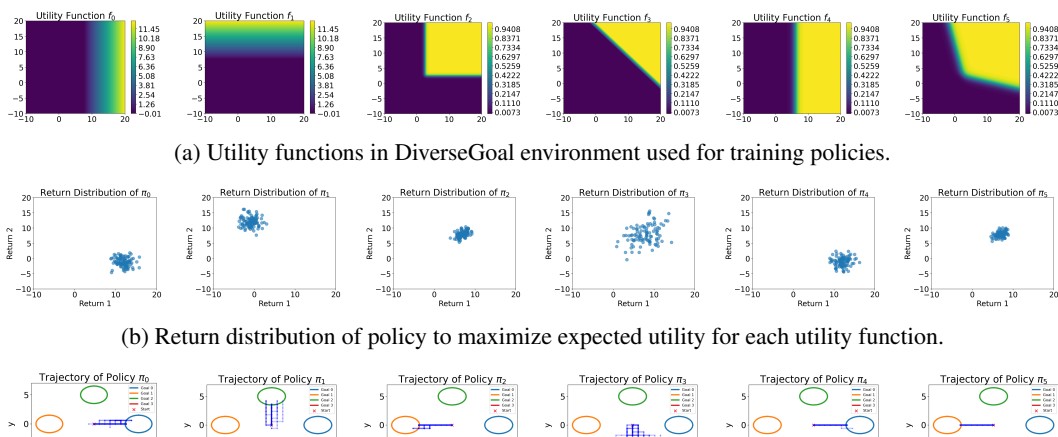

(a) Utility functions in DiverseGoal environment used for training policies.

(b) Return distribution of policy to maximize expected utility for each utility function.

(c) The trajectory of policies learned by maximizing the expected utility for each utility function.

Figure 4: The case study results of DPMORL under DiverseGoal environment.

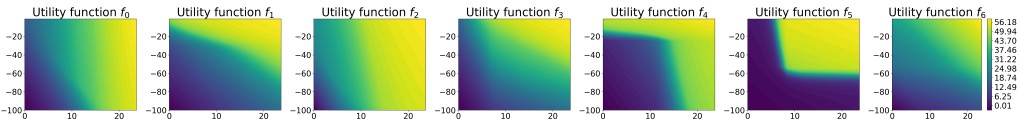

(a) Utility functions generated by our method in Section 4.2.1 under DeepSeaTreasure environment.

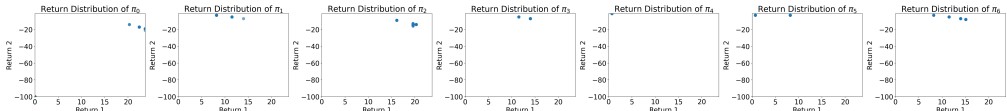

(b) Return distribution of policy learned by DPMORL to maximize expected utility for each utility function.

Figure 5: The case study results of DPMORL under DeepSeaTreasure environment.

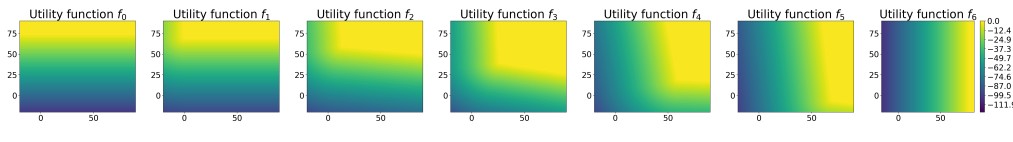

(a) Utility functions with different distributional preferences under Reacher environment.

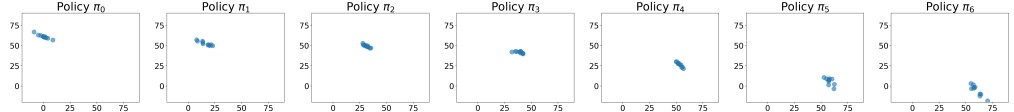

(b) Return distribution of policy learned by DPMORL to maximize expected utility for each utility function.

Figure 6: The case study results of DPMORL under Reacher environment.

with safe RL settings, where the two returns are time penalty and task reward respectively. With our method, users can provide different utility functions derived from safety constraints, and optimize the expected utility via DPMORL for obtaining policies that satisfy the constraint.

## 5.2 Main Experiment

In this section, we illustrate the performance of DPMORL on five environments based on MO-Gymnasium [46] to answer the latter two questions.

**Environments.** We conducted experiments across five environments based on MO-Gymnasium [46] to evaluate the performance of our proposed method, DPMORL. These environments represent a diverse range of tasks, from simple toy problems to more complex continuous control tasks, and cover various aspects of multi-objective reinforcement learning:

- DeepSeaTreasure: A classic MORL benchmark that requires exploration in a gridworld to find treasures with different values and depths.
- FruitTree: A multi-objective variant of the classic gridworld problem, where an agent has to collect different types of fruits with varying rewards and penalties.
- HalCheetah, Hopper, MountainCar: Three continuous control tasks that require controlling different agents for task solving and minimizing energy usage.

More details about the environments are gathered in Appendix C.1.

**Baselines.** We compare DPMORL with four state-of-the-art baselines in the context of distributional preferences of MORL: Optimistic Linear Support (**OLS**) [47, 16]; Prediction-Guided Multi-Objective Reinforcement Learning (**PGMORL**) [8]; Generalized Policy Improvement with Linear Support (**GPI-LS**) [7]; Generalized Policy Improvement with Prioritized Dyna (**GPI-PD**) [7].

**Training Details.** For all methods, including DPMORL and the baselines, each policy was trained for $1 \times 10^7$ steps. We learn a set of $N = 20$ policies for DPMORL and all of the baselines to ensure a fair comparison. Finally, we use the learned $N = 20$ policies in each method for evaluations.

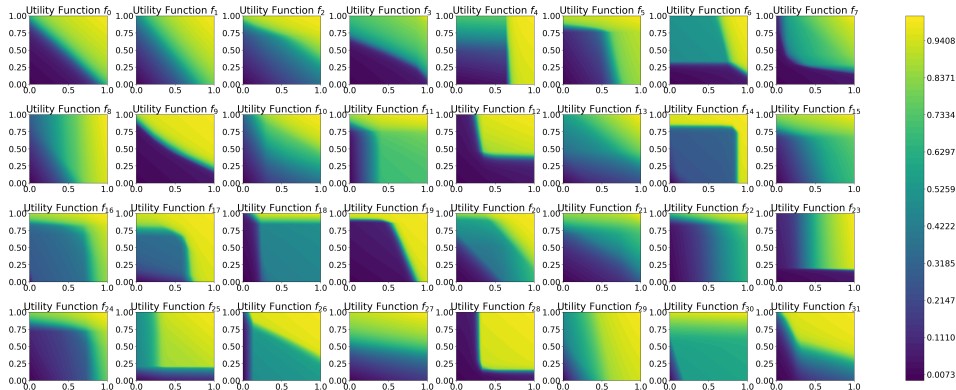

Figure 7: Illustration of 2D utility functions learned by our methods in Section 4.2.1.

**Implementation Details.** We use PPO algorithms implemented in Stable Baselines 3 [48] in Algorithm 1. We use a normalization technique by linearly mapping the return into scale $[0, 1]^K$ without modifying the optimal policies. Detailed implementations are provided in Appendix B.

**Evaluation Metrics.** To thoroughly evaluate the performance of DPMORL and compare it with the baseline methods, we employed four distinct metrics. These comprise two existing MORL metrics, HyperVolume and Expected Utility, in addition to two novel metrics developed specifically for assessing distributional preference, i.e., Constraint Satisfaction and Variance Objective. The latter two are designed to underscore the optimality of multivariate distributions associated with the learned policies. In terms of Constraint Satisfaction, we randomly generate $M = 100$ constraints for the policy set produced. The Constraint Satisfaction metric is then computed by considering the highest probability within the policy set that satisfies each individual constraint. The Variance Objective metric, on the other hand, involves generating $M = 100$ random linear weights. These weights are applied to both the expected returns and the standard deviations of returns in each dimension. This objective encourages attaining greater expected returns while simultaneously reducing variance, thereby catering to dynamic preferences. Further details about the implementation of these evaluation metrics are provided in Appendix C.2 and Appendix C.3.

## 5.3 Results

**Generation of Utility Functions.** In accordance with the methodology detailed in Section 4.2.1, we employ non-decreasing neural networks in conjunction with diversity-focused objectives to generate a diverse assortment of utility functions. The generated functions are visually represented in Figure 7. The outcomes clearly indicate that optimizing diversity-based objective functions allows for generating a broad range of non-linear utility functions, thereby encompassing an expansive array of preferences with respect to returns. Subsequently, we utilize the first $N = 20$ utility functions depicted in Figure 7 to train an equivalent number of policies under DPMORL.

**Standard MORL Metrics.** The results under standard MORL metrics are shown in Table 1. Focusing on the Expected Utility, the performance of DPMORL is the highest in the Hopper, HalfCheetah, MountainCar, and DeepSeaTreasure environments (4/5), indicating that our method outperforms others in terms of expected utility. While for FruitTree environment, DPMORL also obtains consistent performance with the best contender, GPI-LS. In terms of the HyperVolume, DPMORL also performs the best in the Hopper, HalfCheetah, MountainCar, and FruitTree environments (4/5). The results show that DPMORL can yield better performance across both metrics and most environments (Hopper, HalfCheetah, MountainCar, and DeepSeaTreasure), meanwhile delivering robust and consistent results on the other one (FruitTree). Since PGMORL works on continuous action spaces, here we omit the results of PGMORL on environments with discrete action space, DeepSeaTreasure and FruitTree.

**Distributional MORL Metrics.** The results under distributional metrics are shown in Table 2. DPMORL outperforms other methods in most environments on both Constraint Satisfaction (5/5) and Variance Objective (3/5), indicating its strong ability to handle the distributional multi-objective reinforcement learning problem. For the rest environments, DPMORL also obtained comparable results compared with the best contender, GPI-LS. The results show that the policies learned by

Table 1: Experimental results of each method on each standard MORL metric on all five environments. "EU" stands for Expected Utility, and "HV" stands for Hypervolume.

| Environment | Hopper | | HalfCheetah | | MountainCar | | DeepSeaTreasure | | FruitTree | |
| Metric | EU | HV | EU | HV | EU | HV | EU | HV | EU | HV |
|---|---|---|---|---|---|---|---|---|---|---|
| GPI-PD | 2374.44 | 5033458.34 | 412.62 | 1083227.57 | -55.44 | 7367.59 | 5.93 | 9.75 | **6.98** | 0.14 |
| GPI-LS | 1398.00 | 1446705.10 | 98.31 | 2839051.60 | -37.37 | 8052.58 | 5.04 | 9.36 | 3.84 | 1.49 |
| OLS | 175.07 | 3298.13 | -580.38 | 1420270.13 | -470.00 | 2.45 | 4.64 | **10.28** | 6.27 | 7.49 |
| PGMORL | 300.19 | 32621.25 | 111.30 | 383681.25 | -429.48 | 94.45 | - | - | - | - |
| DPMORL (Ours) | **3492.93** | **12154967.99** | **1189.68** | **8593769.62** | **-29.89** | 8663.80 | **6.70** | 8.68 | 6.89 | **16.39** |

Table 2: Experimental results of each method on Distributional metric on all five environments. "Constraints" stands for Constraints Satisfaction, and "Var" stands for Variance Objective.

| Environment | Hopper | | HalfCheetah | | MountainCar | | DeepSeaTreasure | | FruitTree | |
| Metric | Constraints | Var | Constraint | Var | Constraint | Var | Constraint | Var | Constraint | Var |
|---|---|---|---|---|---|---|---|---|---|---|
| GPI-PD | 0.47 | 979.74 | 0.64 | 83.49 | **1.00** | -31.15 | 0.85 | **2.59** | 0.65 | **3.42** |
| GPI-LS | 0.25 | 607.47 | 0.60 | 51.67 | **1.00** | -22.73 | 0.85 | 1.99 | 0.33 | 1.35 |
| OLS | 0.05 | 75.48 | 0.43 | -311.27 | 0.05 | -244.21 | 0.80 | 1.86 | 0.50 | 2.98 |
| PGMORL | 0.05 | 98.47 | 0.50 | 45.48 | 0.11 | -249.69 | - | - | - | - |
| DPMORL (Ours) | **0.76** | **1645.89** | **0.82** | **431.26** | **1.00** | **-16.32** | **0.90** | 2.54 | **0.67** | 3.21 |

DPMORL have a higher probability of satisfying randomly generated constraints, and can better balance the trade-off between expectations and variances.

**Return Distributions of Learned Policies by DPMORL.** We provide visualizations of the multi-dimensional return distributions of each policy learned by DPMORL and four baseline methods (PGMORL, OLS, GPI-LS and GPI-PD) on different MORL environments, as shown in Appendix C.4. DPMORL's learned policy set demonstrates higher diversity with respect to return distributions compared to baseline methods.

**DPMORL on More Than 2-dimensional Return Space.** We run experiments with 3 objectives under Hopper and FruitTree environment, and show the learned return distribution and standard MORL evaluation metrics in Appendix C.5. The results demonstrate that DPMORL can effectively learn DPO policies under more than 3-dimensional reward functions, and achieves better performance in different MORL metrics compared to baseline methods under 3-dimensional reward functions.

**Ablation Analysis with Variable Number of Policies.** We provide the ablation studies of DPMORL with a varying amount of learned policies $N$ in Appendix C.6. As the number of policies increases, both expected utility and hypervolume increase, which means DPMORL can indeed obtain better performance as the number of policies increases. On the other hand, DPMORL has obtained promising results when there exist only 5 policies, which verify the effectiveness of DPMORL in all environments.

## 6   Conclusion

In this work, we initialized the study of distributional MORL, specifically when preferences over different objectives and their return distributions are uncertain. We introduced the concept of Distributional Pareto-Optimal (DPO) policies with rigorous theoretical analysis, which extend the notion of Pareto-optimality in MORL to include preferences over the entire distribution of returns, not just their expected values. To obtain such desirable policies, we proposed a new algorithm, DPMORL, designed to learn DPO policies with non-linear utility functions. DPMORL allows for expressing a wide range of distributional preferences, providing a flexible and expressive approach to MORL. Experiment results showed that DPMORL consistently outperformed existing MORL methods in terms of optimizing policies for multivariate expected and distributional preferences.

## Acknowledgments

This research was supported by the Institute for AI and Beyond, UTokyo; JST SPRING, Grant Number JPMJSP2108; and Microsoft Research Asia D-CORE program. The authors would like to thank Soichiro Nishimori, Kazuki Ota, and the anonymous reviewers for their insightful comments and suggestions.

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

# A Proof

**Lemma 1.** *If a policy $\pi$ is the only optimal policy under some non-decreasing utility function $f$, then it is a Distributionally Pareto-Optimal policy.*

*Proof.* If $\pi$ is the only optimal policy under non-decreasing utility function $f$, then for any policy policy $\pi'$,

$$\mathbb{E}_{\boldsymbol{z}\sim\boldsymbol{\mu}(\pi)}f(\boldsymbol{z}) > \mathbb{E}_{\boldsymbol{z}\sim\boldsymbol{\mu}(\pi')}f(\boldsymbol{z}). \tag{4}$$

From Definition 1 and Definition 2, if any policy $\pi'$ stochastically dominates $\pi$, then for any utility function $f$, $\mathbb{E}_{\boldsymbol{z}\sim\boldsymbol{\mu}(\pi')} \geq \mathbb{E}_{\boldsymbol{z}\sim\boldsymbol{\mu}(\pi)}f(\boldsymbol{z})$, which makes a contradition. Therefore, $\pi$ is not stochastically dominated by any other policies, and $\pi$ is a Distributional Pareto-Optimal policy. $\square$

**Lemma 2.** *If a policy $\pi$ is one of the optimal policy under some strictly increasing utility function $f$, then it is a Distributionally Pareto-Optimal policy. The utility function $f$ is strictly increasing if and only if $x \succeq y$ and $x \neq y \Rightarrow f(x) > f(y)$.*

*Proof.* Following the Equation (2) and (3) of [49] which is derived from Kantorovich duality (Theorem 5.10 in [50]), we have

$$\boldsymbol{\mu}_1 \succ_{SD} \boldsymbol{\mu}_2 \text{ if and only if } \boldsymbol{\mu}_1 \neq \boldsymbol{\mu}_2 \text{ and } \max_{\gamma \in \Pi(\boldsymbol{\mu}_1, \boldsymbol{\mu}_2)} \mathbb{E}_{(x,y)\sim\gamma}[\mathbb{1}_{x \succeq y}] = 1 \tag{5}$$

where $\Pi(\boldsymbol{\mu}_1, \boldsymbol{\mu}_2)$ is the set of all couplings of $\boldsymbol{\mu}_1$ and $\boldsymbol{\mu}_2$, which is the set of distributions $\boldsymbol{\gamma}(x, y)$ such that

$$\int \boldsymbol{\gamma}(x, y)dy = \boldsymbol{\mu}_1(x), \int \boldsymbol{\gamma}(x, y)dx = \boldsymbol{\mu}_2(y).$$

Suppose there exists a policy $\pi'$ stochastically dominate $\pi$, then there exists $\boldsymbol{\gamma}(x, y) \in \Pi(\boldsymbol{\mu}(\pi'), \boldsymbol{\mu}(\pi))$, such that $\mathbb{E}_{(x,y)\sim\gamma}[\mathbb{1}_{x \succeq y}] = 1$. Since $\pi$ is one of the optimal policy under strictly increasing utility function $f$, we have

$$\mathbb{E}_{x\sim\boldsymbol{\mu}(\pi')}f(x) - \mathbb{E}_{y\sim\boldsymbol{\mu}(\pi)}f(y) \leq 0$$

$$\int f(x)\boldsymbol{\mu}(\pi')(x)dx - \int f(y)\boldsymbol{\mu}(\pi)(y)dy \leq 0$$

$$\iint [f(x) - f(y)]\boldsymbol{\gamma}(x, y)dxdy \leq 0$$

Since $\mathbb{E}_{(x,y)\sim\gamma}[\mathbb{1}_{x \succeq y}] = 1$, i.e. $\iint_{x \succeq y} \boldsymbol{\gamma}(x, y)dxdy = 1$, we have

$$\iint_{x \not\succeq y} [f(x) - f(y)]\boldsymbol{\gamma}(x, y)dxdy = 0$$

and

$$\iint_{x \succeq y} [f(x) - f(y)]\boldsymbol{\gamma}(x, y)dxdy \leq 0$$

There are two cases. Case 1: there exists $c \succ 0$ such that $\iint_{x \succeq y+c} \boldsymbol{\gamma}(x, y)dxdy > 0$.

Since $f$ is an strictly increasing function, we have

$$\text{If } x \succeq y \text{ and } x \neq y, \text{ then } f(x) > f(y).$$

Then, we have:

$$0 \geq \iint_{x \succeq y} [f(x) - f(y)]\boldsymbol{\gamma}(x, y)dxdy$$

$$\geq \iint_{x \succeq y+c} [f(x) - f(y)]\boldsymbol{\gamma}(x, y)dxdy + \iint_{y+c \succ x \succeq y} [f(x) - f(y)]\boldsymbol{\gamma}(x, y)dxdy$$

$$\geq \iint_{x \succeq y+c} [f(y + c) - f(y)]\boldsymbol{\gamma}(x, y)dxdy$$

Since $c \succ 0$, $f(y+c) - f(y) > 0$. Since $\iint_{x \succeq y+c} \boldsymbol{\gamma}(x,y) > 0$ and $\boldsymbol{\gamma}(x,y)$ is continuous almost everywhere, we have $\iint_{x \succeq y+c} [f(y+c) - f(y)] \boldsymbol{\gamma}(x,y) dx dy > 0$, which yields a contradiction.

Case 2: there does not $c \succ 0$ such that $\iint_{x \succeq y+c} \boldsymbol{\gamma}(x,y) dx dy > 0$. Then, we have $\mathbb{E}_{(x,y)\sim\gamma(x,y)} \mathbb{1}[x = y] = 1$, which contradicts with the assuption that $\mu(\pi') \neq \mu(\pi)$.

Therefore, we have proved that there does not exist $\pi'$ that stochastically dominate $\pi$, when $\pi$ has optimal expected utility under some strictly increasing utility function $f$. $\qquad\square$

**Lemma 3.** *Any Pareto-Optimal policy is a Distributionally Pareto-Optimal policy.*

*Proof.* We first show that if distribution $\boldsymbol{\mu}'$ stochastically dominates $\boldsymbol{\mu}$, then the expectation $\mathbb{E}_{\boldsymbol{z}\sim\boldsymbol{\mu}'}[\boldsymbol{z}]$ dominates $\mathbb{E}_{\boldsymbol{z}\sim\boldsymbol{\mu}}[\boldsymbol{z}]$. Following Kantorovich duality, there exists a coupling $\gamma$ of $\boldsymbol{\mu}'$ and $\boldsymbol{\mu}$ that $\mathbb{E}_{(x,y)\sim\gamma}[\mathbb{1}_{x \succeq y}] = 1$. Then,

$$\mathbb{E}_{\boldsymbol{z}\sim\boldsymbol{\mu}'}[\boldsymbol{z}] - \mathbb{E}_{\boldsymbol{z}\sim\boldsymbol{\mu}}[\boldsymbol{z}]$$
$$= \mathbb{E}_{(x,y)\sim\gamma}[x - y]$$
$$\succeq \mathbb{E}_{(x,y)\sim\gamma}[\boldsymbol{0}] = \boldsymbol{0}$$

When $\mathbb{E}_{(x,y)\sim\gamma}[x - y] = \boldsymbol{0}$, it requires that $x = y$ holds where $\gamma(x,y) > 0$, which makes $\boldsymbol{\mu}' = \boldsymbol{\mu}$ that contradicts with the fact that $\boldsymbol{\mu}' \succ_{SD} \boldsymbol{\mu}$. Therefore, the expectation $\mathbb{E}_{\boldsymbol{z}\sim\boldsymbol{\mu}'}[\boldsymbol{z}]$ dominates the expectation $\mathbb{E}_{\boldsymbol{z}\sim\boldsymbol{\mu}}[\boldsymbol{z}]$. It follows that if the expectation $\mathbb{E}_{\boldsymbol{z}\sim\boldsymbol{\mu}}[\boldsymbol{z}]$ is not dominated by the expectation $\mathbb{E}_{\boldsymbol{z}\sim\boldsymbol{\mu}'}[\boldsymbol{z}]$, then the distribution $\boldsymbol{\mu}$ is not stochastically dominated by $\boldsymbol{\mu}'$.

If policy $\pi$ is Pareto-optimal, then its multidimensional expected returns $\mathbb{E}[\boldsymbol{Z}(\pi)]$ are not dominated by $\mathbb{E}[\boldsymbol{Z}(\pi')]$ of any other policies $\pi'$. Then, the return distribution $\boldsymbol{\mu}(\pi)$ of policy $\pi$ is not stochastically dominated by the return distribution $\boldsymbol{\mu}(\pi')$ of any other policies $\pi'$. Therefore, policy $\pi$ is a Distributionally Pareto-Optimal policy. To conclude, any Pareto-Optimal policy is a Distributionally Pareto-Optimal policy. $\qquad\square$

**Theorem 1.** *If a policy $\pi$ has optimal expected utility under some non-decreasing utility function $f$, where either*

  1. *utility function $f$ is strictly increasing*

  2. *$\pi$ is the only optimal policy with greatest expected utility under utility function $f$*

*then $\pi$ is a Distributionally Pareto-Optimal policy. Also, any Pareto-Optimal policy is a Distributionally Pareto-Optimal policy.*

*Proof.* The theorem directly follows from Lemma 1, Lemma 2 and Lemma 3. $\qquad\square$

The following theorem proves that the optimal policy over Value at Risk (VaR) and over Conditional Value at Risk (CVaR) belong to the set of Distributionally Pareto-Optimal policies. Here, we use the random variable $\boldsymbol{\mu}(\pi)$ to represent the negative risk, where smaller value represents larger risks, in order to be aligned with the definition of Pareto-Optimality.

**Definition 4.** *Given a distribution $\mu$, the Value at Risk (VaR) is defined as*

$$VaR_\alpha(\mu) = \min\{z_0 | \mathbb{E}_{z\sim\mu} \mathbb{1}[z \leq z_0] \geq \alpha\}$$

**Definition 5.** *Given a distribution $\mu$, the Conditional Value at Risk (CVaR) is defined as*

$$CVaR_\alpha(\mu) = \mathbb{E}_{z\sim\mu}[z | z \leq VaR_\alpha(\mu)]$$

**Theorem 3.** *When $K = 1$, there exists policy with optimal Value at Risk (VaR), and policy with optimal Condition Value at Risk (CVaR), which belongs to the set of Distributional Pareto-Optimal Policies.*

*Proof.* We let the policy set $\Pi_{\text{VaR}}$ as the policy set with optimal value at risk, which $z_0 = VaR_\alpha(\mu(\pi))$ for any $\pi \in \Pi_{\text{VaR}}$. We let $\Pi'_{\text{VaR}}$ to be the policy set with non-optimal policy, with $VaR_\alpha(\mu(\pi')) < z_0$ for any $\pi' \in \Pi'_{\text{VaR}}$. Suppose all Distributionally Pareto-Optimal policy belongs to $\Pi'_{\text{VaR}}$. We define a

utility function as $f(z) = \begin{cases} 1, & z \geq z_0 \\ 0, & z < z_0 \end{cases}$, which is non-decreasing. Then, we have $\mathbb{E}_{z \sim \mu(\pi)} f(z) \geq \alpha$ for any $\pi \in \Pi_{\text{VaR}}$, while $\mathbb{E}_{z \sim \mu(\pi')} f(z) < \alpha$ for any $\pi' \in \Pi'_{\text{VaR}}$. Therefore, any policy $\pi \in \Pi_{\text{VaR}}$ is not stochastically dominated by any policy $\pi' \in \Pi'_{\text{VaR}}$. Therefore, there must exists a Distributional Pareto-Optimal Policy in $\Pi$, which results in a contradiction. Therefore, there must exists a policy with optimal value at risk, which is Distributionally Pareto-Optimal.

Similarly to the case in VaR, we let the policy set $\Pi_{\text{CVaR}}$ as the policy set with optimal conditional value at risk, which $z_0 = \text{CVaR}_\alpha(\mu(\pi))$ for any $\pi \in \Pi_{\text{CVaR}}$. We let $\Pi'_{\text{CVaR}}$ to be the policy set with non-optimal CVaR, with $\text{CVaR}_\alpha(\mu(\pi')) < z_0$ for any $\pi' \in \Pi'_{\text{CVaR}}$. Then we can prove that any policy $\Pi'_{\text{CVaR}}$ does not stocastically dominate the policy in $\Pi_{\text{CVaR}}$. Suppose the opposite case where $\pi \in \Pi_{\text{CVaR}}$ is dominated by $\pi' \in \Pi'_{\text{CVaR}}$. We let $f(z), f'(z)$ to be the cumulative distribution function of $\mu(\pi)$ and $\mu(\pi')$, and $f(z) \geq f'(z)$ must be satisfied to make $\pi \prec_{SD} \pi'$. Therefore,

$$z_0 = \text{CVaR}(\mu(\pi)) = \mathbb{E}_{z \sim \mathcal{U}(0,\alpha)} f^{-1}(z) \leq \mathbb{E}_{z \sim \mathcal{U}(0,\alpha)} f'^{-1}(z) = \text{CVaR}(\mu(\pi'))$$

which results in a contradiction. Therefore, any policy $\Pi'_{\text{CVaR}}$ does not stocastically dominate the policy in $\Pi_{\text{CVaR}}$. We thus proved that there exists a Distributionally Pareto-Optimal policy in $\Pi_{\text{CVaR}}$, which is an optimal policy under conditional value at risk.

$\square$

## B  Implementation Details

**Non-Decreasing Neural Network.**  We used Non-Decreasing Neural Network to parameterize the utility function. The network architecture consisted of three fully connected layers, with our designed non-decreasing activation activation functions applied to the hidden layers:

$$f(x) = \text{concat}(max(x, -0.5), min(x, 0.5), clip(x, -0.5, 0.5))$$

This design of activation function is an extension to ReLU activation function, which empirically make the learned utility function more diverse, covering more types of decision boundaries. After each gradient update during training, we clip all the trainable weights in the network to be non-negative to ensure that the resulting utility function was non-decreasing. The network's input dimension corresponded to the number of objectives in the environment, and the output dimension was a scalar representing the utility value.

**Return Normalization.**  We use a normalization technique by linearly mapping the return into scale $[0, 1]^K$ without modifying the optimal policies. Specifically, we keep track of the minimum and maximum multivariate returns that the policies have currently achieved, denote by $z_{min}$ and $z_{max}$. We let $z_{mid} = (z_{min} + z_{max})/2$, and $d = \max_i[z_{max,i} - z_{min,i}]$. Before the any multivariate return $z$ is fed into the utility function, we normalize it by

$$z_{norm} = \frac{1}{d}(z_{norm} - z_{mid}) + \frac{1}{2}$$

This maps $z$ linearly to $z_{norm}$, where $z_{norm} \in [0, 1]^K$, without affecting the scale of each dimension of returns. In Algorithm 1, the return $z_t$ is normalized before feeding into the utility function to compute the scalar reward.

**DPMORL Algorithm.**  For the implementation of our DPMORL algorithm, we used the Proximal Policy Optimization (PPO) algorithm [44] as the basic RL algorithm. The learning rate was set to $3 \times 10^{-4}$, with a discount factor $\gamma$ of 0.99. For PPO, we used a clipping parameter of 0.2. The batch size was set to 256 for all environments and algorithms, with updates performed every 2,048 steps. The utility functions were trained using the Adam optimizer [51], with a learning rate of $3 \times 10^{-4}$.

## C  Experimental Details

### C.1  Environments

The details of the environments used in the main paper are listed below:

- DeepSeaTreasure: A classic MORL benchmark that requires exploration in a gridworld to find treasures with different values and depths.
- FruitTree: A multi-objective variant of the classic gridworld problem, where an agent has to collect different types of fruits with varying rewards and penalties.
- HalCheetah: A continuous control task where a simulated cheetah must run and jump over obstacles while minimizing energy consumption.
- Hopper: A continuous control task that involves balancing and hopping a simulated one-legged robot while minimizing energy usage.
- MountainCar: A classic reinforcement learning problem, extended to include multiple objectives, such as minimizing the time to reach the goal and reducing energy consumption.

In the Appendix, we additionally add one environment:

- Reacher: A discrete control task where a simulated robotic arm must reach a target while minimizing energy usage.

## C.2 Standard MORL Metrics

The details of the standard MORL metrics used in the main paper are listed below:

- Expected Utility is similar to R-Metrics used in Multi-Objective Optimization (MOO). This metric is concerned with the expected utility of the policies on the Pareto front for various weights. It provides a means to assess the quality of the policies derived by the algorithm in terms of their expected performance across different utility functions. A higher Expected Utility indicates that the policies offer a better trade-off between the objectives, thereby leading to more desirable outcomes under a variety of user preferences:

$$\text{EU}(\Pi) = \mathbb{E}_{\mathbf{w} \sim \mathcal{W}}[\max_{\pi \in \Pi} f_{\mathbf{w}}(\mathbf{z})], \tag{6}$$

where $f_{\mathbf{w}}$ denotes the user's utility function linearly parameterized by $\mathbf{w}$, and $\mathbf{z}$ denotes the returns of one episode.

- Hypervolume is a prominent measure used in Multi-Objective Optimization (MOO) and Multi-Objective Reinforcement Learning (MORL). This metric calculates the volume in the objective space dominated by the Pareto front with respect to a reference point. The Hypervolume gives a scalar measure of the quality of a Pareto front, with higher values indicating a better Pareto front. A Pareto front with a larger Hypervolume signifies that the algorithm can yield a wider range of policies, thus dominating more of the objective space, which indicates a better exploration of the policy space:

$$\text{HV}(\Pi, r) = \text{Leb}(\mathbf{y} \in \mathbb{R}^K \mid \exists \pi \in \Pi : \mathbf{z} \prec_{SD} \mathbf{y} \prec_{SD} \mathbf{r}), \tag{7}$$

where $r$ is the reference point in the objective space and Leb denotes the Lebesgue measure, which in this context refers to the volume in the objective space. The reference points used in the experiment of the main paper are calculated as the minimal mean returns of each method under evaluation.

## C.3 Distributional MORL Metrics

We propose two novel metrics for assessing distributional preference: Constraint Satisfaction and Variance Objective.

For Constraint Satisfaction, we define $u(p, \pi)$ to be the probability of return random variance $z \sim \mathbf{Z}(\pi)$ that satisfy the constraint $p$, representing how a policy satisfy the given constraint. We randomly generate each constraint defined by $n$ linear constraints represented by $\{\mathbf{z} | \forall i, \mathbf{w}_i^T \mathbf{z} \geq c_i\}$, where $w_i$ is uniformly sampled from the surface $\mathbf{w}_i \succeq 0, \mathbf{1}^T \mathbf{w}_i = 1$, and $c_i$ is sampled from the distribution of $\mathcal{U}(\min_{\mathbf{z} \in \mathcal{Z}} \mathbf{w}_i^T \mathbf{z}, \max_{\mathbf{z} \in \mathcal{Z}} \mathbf{w}_i^T \mathbf{z})$, where $\mathcal{Z}$ is the set of all return samples of the final policies learned by DPMORL.

For Variance Objective, we define $u(p, \pi)$ to be a weighted sum of expected return $\mathbb{E}_{\mathbf{z} \sim \boldsymbol{\mu}(\pi)} \mathbf{z}$ and standard deviation of return $\sqrt{\text{Var}_{\mathbf{z} \sim \boldsymbol{\mu}(\pi)}[\mathbf{z}]}$. We randomly generate $\mathbf{w}_i \in \mathbb{R}^{2K}$, where $\mathbf{w}_i \succeq 0$ and

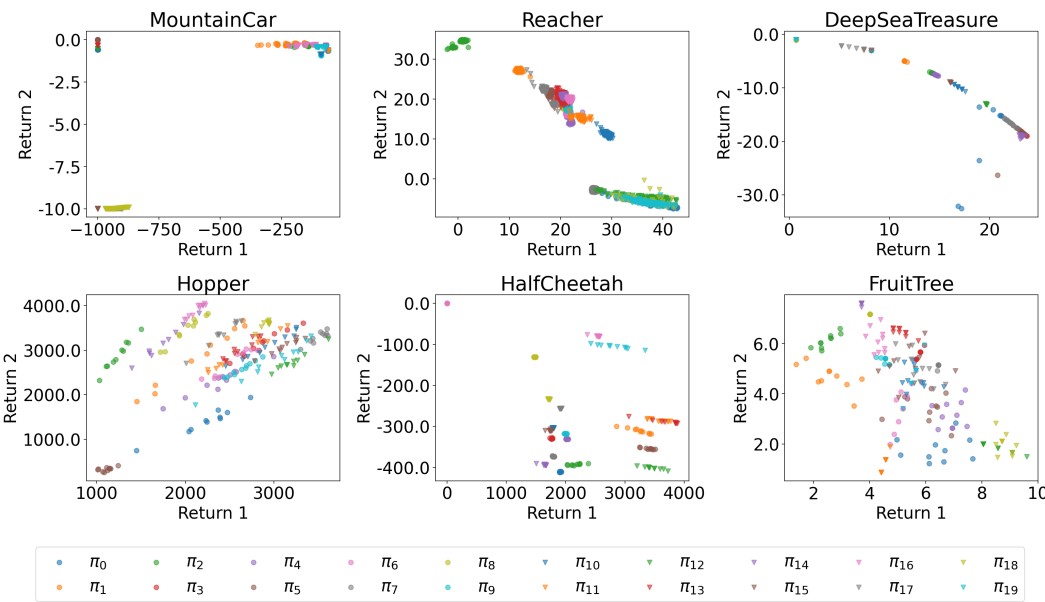

Figure 8: Return distribution of the set of policies learned by DPMORL on each environment.

$\mathbf{1}^T \boldsymbol{w}_i = 1$. We let $\boldsymbol{w}_i = [\boldsymbol{w}_i^1, \boldsymbol{w}_i^2]$, where $\boldsymbol{w}_i^1, \boldsymbol{w}_i^2 \in \mathbb{R}^K$, and use $\boldsymbol{w}_i^1$ and $\text{-}\boldsymbol{w}_i^2$ as the weight for expected return and standard derivation of return respectively.

For both of the metric, we define $u(p, \pi)$ to be the satisfaction score of policy $\pi$ under preference $p$. We generate $M = 100$ number of preference $p_1, p_2, \cdots, p_M$. For both of the Constraint Satisfaction and Variance Objective metric, the mean score is defined as

$$\frac{1}{M} \sum_{i=1}^{M} \max_{j \in \{1, 2, \cdots, N\}} u(p_i, \pi_j)$$

The mean score in the above equation measures the best performance of satisfying the preference of the user, averaged in $M = 100$ preferences. Higher mean score represents better ability of meeting different preferences. In Table 2 of the main experiment, all of the numbers are calculated with the above equation.

## C.4   Return Distributions of Learned Policies by DPMORL and Baseline Methods

We demonstrate the return distribution of the learned policies by DPMORL in Figure 8, which demonstrates that DPMORL is able to learn a set of policies with optimal return distributions. We also visualize the return distribution of policies learned by four baseline methods, which is shown in Figure 9. DPMORL's learned policy set has higher diversity with respect to return distributions compared to baseline methods.

## C.5   DPMORL on More Than 2-Dimensional Return Space

We run experiments with 3 objectives under Hopper and FruitTree environment, where we use the first three dimension of the reward in the MO-Hopper and Fruit-Tree environment the MO-Gymnasium. We show the 3-dimensional return distribution of the set of policies learned by DPMORL and baseline methods in Figure 10. The results demonstrate that DPMORL can effectively learn DPO policies under more than two-dimensional reward functions. Also, DPMORL learns better 3-dimensional return distributions than GPI-PD, a competitive MORL baseline method. We also show quantitative evaluation metrics under Hopper and FruitTree environments with 3 objectives in Table 3. The results demonstrate that DPMORL achieves better performance in different MORL metrics compared to baseline methods under 3-dimensional reward functions.

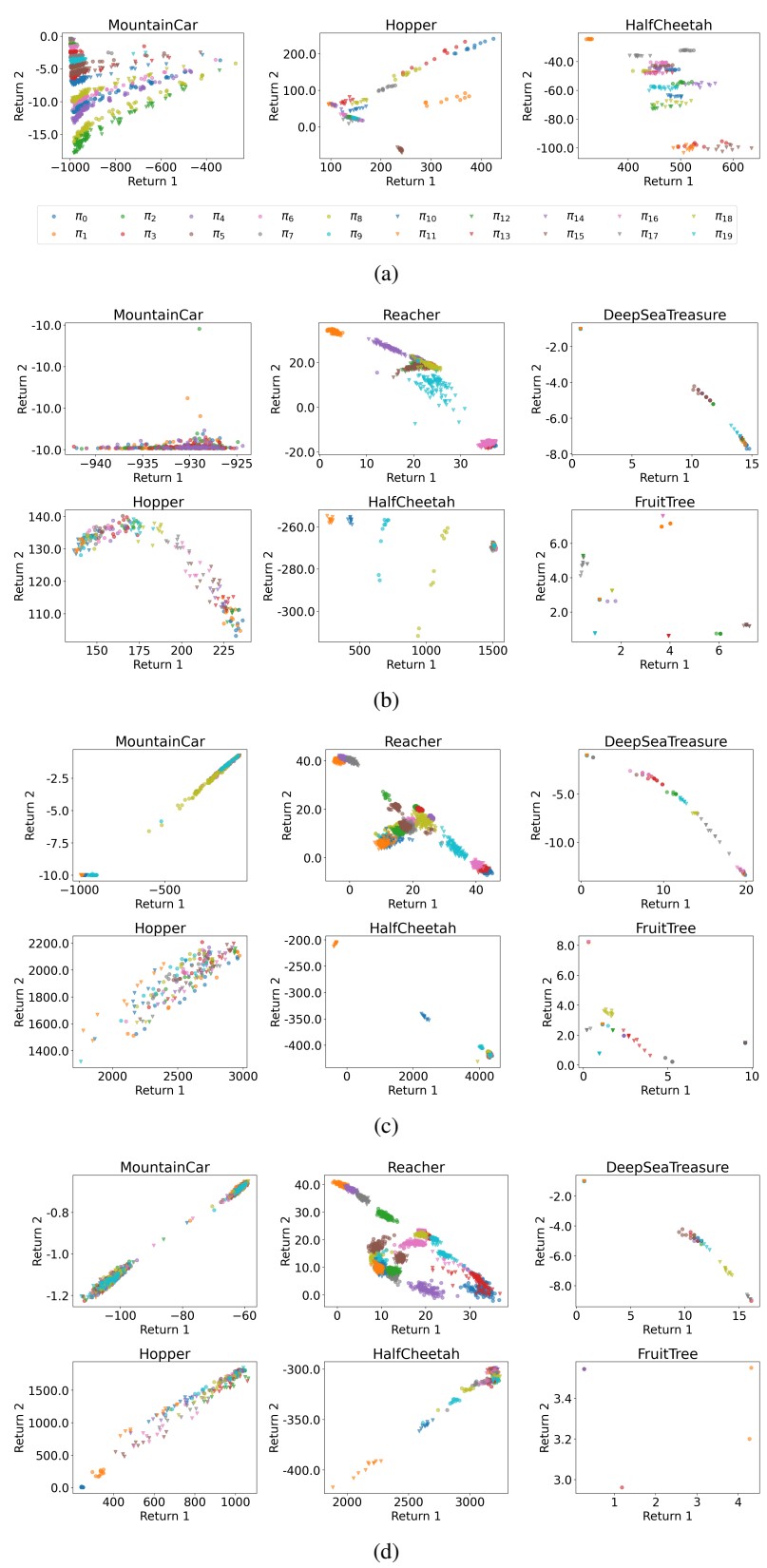

Figure 9: Return distribution of the policy set learned by four baseline methods: PGMORL, OLS, GPI-LS, and GPI-PD.

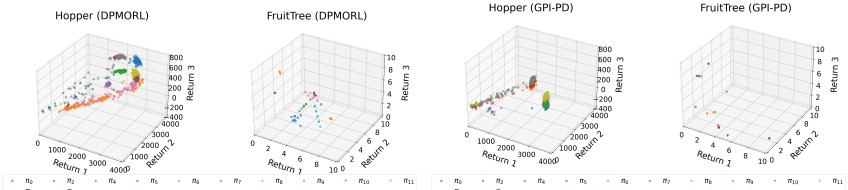

Figure 10: DPMORL can effectively learn DPO policies under more than two-dimensional reward functions. Left: 3-dimensional return distribution of the set of policies learned by DPMORL under Hopper and FruitTree environment. Right: 3-dimensional return distribution under the same environment learned by GPI-PD.

Table 3: Experimental results of each method on Hopper and FruitTree environments with 3-dimensional reward function.

| Environment | Hopper | | | | FruitTree | | | |
|---|---|---|---|---|---|---|---|---|
| Metric | Expected Utility | CVaR | Constraint Satisfaction | Variance Objective | Expected Utility | CVaR | Constraint Satisfaction | Variance Objective |
| GPI-LS | 1037.77 | 1286.61 | 0.15 | 480.94 | 4.19 | 4.55 | 0.27 | 1.60 |
| GPI-PD | 1511.40 | 1339.67 | 0.36 | 664.02 | 4.44 | 6.76 | 0.20 | 2.30 |
| OLS | 441.62 | 194.56 | 0.08 | 78.77 | 4.72 | 6.96 | 0.35 | 2.36 |
| DPMORL | **2107.26** | **2840.44** | **0.67** | **949.96** | **5.32** | **7.25** | **0.38** | **2.57** |

## C.6 Ablation Analysis with Variable Number of Policies

We also provide the ablation studies of DPMORL with a varying amount of learned policies $N$, as shown in Table 4. We can see that as the number of policies increases, both expected utility and hypervolume increase, which indicates that DPMORL can indeed obtain better performance as the number of policies increases. On the other hand, DPMORL has obtained promising results when there exist only 5 policies, which verify the effectiveness of DPMORL in all environments.

Table 4: Experimental results of DPMORL on each standard MORL metric on five environments with various number of policies. "EU" stands for Expected Utility, and "HV" stands for Hypervolume.

| Environment | Hopper | | HalfCheetah | | MountainCar | | DeepSeaTreasure | | FruitTree | |
|---|---|---|---|---|---|---|---|---|---|---|
| Metric | EU | HV | EU | HV | EU | HV | EU | HV | EU | HV |
| DPMORL ($N = 5$) | 3274.36 | 9917663.14 | 817.61 | 5676147.04 | -29.94 | 7462.10 | 5.82 | 8.93 | 6.22 | 18.74 |
| DPMORL ($N = 10$) | 3309.52 | 10512291.02 | 1244.48 | 8497055.37 | -29.94 | 7465.59 | 5.86 | 14.88 | 6.65 | 21.64 |
| DPMORL ($N = 15$) | 3377.75 | 11453079.82 | 1245.63 | 8497055.37 | -29.94 | 7465.59 | 5.90 | 15.02 | 6.68 | 23.80 |
| DPMORL ($N = 20$) | 3377.75 | 11453079.82 | 1749.27 | 13417685.27 | -29.94 | 7465.59 | 6.53 | 19.68 | 6.68 | 23.80 |

