# OpenReview forum: "Distributional Pareto-Optimal Multi-Objective Reinforcement Learning"
_NeurIPS.cc/2023/Conference — NeurIPS 2023 poster_

### Official Review · Reviewer_1W6z · 2023-07-05

**Soundness:** 3 good
**Presentation:** 4 excellent
**Contribution:** 3 good
**Rating:** 8
**Confidence:** 5

**Summary:**

The paper introduces a new approach for solving Multi-Objective Reinforcement Learning (MORL) problems. Traditional MORL methods aim at optimizing multiple objectives, but generally focus on the expected values of returns, which can be inadequate in real-world scenarios with diverse preferences over returns. The paper addresses this limitation by extending Pareto-optimality to include distributional preferences, called Distributional Pareto-Optimality (DPO). The authors also include discussions of the relationship between Pareto-optimality policies and Distributional Pareto-Optimality policies. Furthermore, the authors propose a novel algorithm, DPMORL, that learns policies considering both the return distributions and their expectations. It captures the optimality of multivariate distributions through stochastic dominance. Through experiments on several benchmarks, DPMORL was found to be effective in learning distributional Pareto-optimal policies and outperformed existing MORL methods.

**Strengths:**

The paper astutely addresses a significant perspective in current MORL research by emphasizing the importance of not only accounting for the expected values of different objectives but also examining the distributional properties of returns in the context of the heterogeneity and diversity inherent in users' preferences. DPMORL stands out for its remarkable flexibility, as it adeptly accommodates a wide spectrum of distributional preferences, making it highly versatile and well-suited for an array of problem scenarios. This attributes to a more nuanced and expressive approach to MORL. Moreover, the algorithm is anchored in a robust theoretical foundation. The paper lucidly delineates Distributional Pareto-Optimal policies and astutely establishes their interrelation with utility functions. This amalgamation of theoretical rigor and empirical validation accentuates the potential of DPMORL to tackle distributional MORL challenges with efficacy. Furthermore, the authors deserve commendation for the inclusion of an intuitive and easily comprehensible case study that effectively demonstrates the utility function of learning.

**Weaknesses:**

The figures of illustration of 2D utility functions, including the ones in Appendix, are arranged somewhat randomly. It is suggested that the authors arrange the results of the utility function according to some gradual pattern.

**Questions:**

I would appreciate further clarification regarding Table 2. Specifically, what are the constraints being referenced in this table? Could you provide illustrative examples to enhance understanding?

**Limitations:**

It would be better to delve deeper into the distribution properties of preferences/constraints and to illuminate the limitations therein with greater specificity. Including illustrative examples of such constraints would not only bolster comprehension but also offer valuable insights into the practical implications and considerations of employing DPMORL in real-world scenarios.

---

> ### Author Rebuttal · Authors · 2023-08-10
>
> To Reviewer 1W6z:
> Thanks for your valuable feedback! We provide the response to each of your questions as follows.
>
> **Q1:** “I would appreciate further clarification regarding Table 2. Especially, what are the constraints being referenced in this table? Could you provide illustrative examples to enhance understanding?”
>
> **A1:** The way of constructing the constraints in Table 2 is detailed in the “Evaluation Metrics” paragraph of Appendix B. Briefly, each constraint is a combination of multiple linear constraints for the multidimensional returns. This covers different constraint satisfaction scenarios, including single safety constraints and multiple combinatorial safety constraints. We will elucidate this with specific examples in our revised manuscript.
>
> **Q2:** “It is suggested that the authors arrange the results of the utility function according to some gradual pattern.”
>
> **A2:** We appreciate your feedback. Our initial intent was to portray the diversity of utility functions, hence the randomized arrangement. Based on your suggestion, we will arrange the utility functions by their average slopes for better clarity in the revised version.

---

> > ### Comment · Reviewer_1W6z · 2023-08-20
> >
> > I thank the authors for providing additional experiments and discussions. I have also read the authors' responses to other reviewers' comments. I think the authors have successfully addressed the questions and concerns raised by other reviewers in the rebuttal statement and additional experimental results. I believe this work represents a valuable contribution to the field of MORL. Therefore, I stand by my previous recommendation to accept this paper at NeurIPS 2023.

---

> > > ### Author Response · Authors · 2023-08-21
> > > **Thank you**
> > >
> > > We appreciate the reviewer's constructive feedback and encouraging comments. In response to your suggestions, we plan to enhance our paper by incorporating examples of constraints. Additionally, we would be more than happy to address any further questions or concerns.

---

### Official Review · Reviewer_GTy6 · 2023-07-07

**Soundness:** 3 good
**Presentation:** 3 good
**Contribution:** 2 fair
**Rating:** 5
**Confidence:** 4

**Summary:**

The authors propose extending multi-objective RL (MORL) to policies that are Pareto optimal over distributions of non-negative utility functions. Specifically, they define distributional Pareto optimal (DPO) policies as those whose expected returns are non-dominated for any non-negative utility function. They then propose learning a set of DPO policies by first learning a set of diverse non-negative utility functions over $[0 ~ 1]^K$ bounded objectives, then learning their corresponding optimal policies using PPO. They demonstrate in several experiments that their approach outperforms prior works in both hyper-volume and expected utility of the learned policy set.

**Strengths:**

- The paper is very well written, clearly presented, and tackles the very significant problem of finding policies that are distributionally Pareto-optimal in MORL.

- I like the paper's formal treatment of Distributional MORL. The definitions and theorems given are particularly clear and precise, showing that the proposed framework is sound (not including how to actually find DPO policies, and not saying I agree with said definitions).

- The proposed approach for finding DPO policies is particularly interesting. By learning a set of utility functions using a diversity metric, then converting the learned utilities to a reward function, one can then use any off-the-shelf RL algorithm to learn the DPO policies. The paper then shows that these optimal policies are guaranteed to maximise their respective utilities.

- The paper conducts several experiments demonstrating that the learned utility functions are indeed diverse and non-decreasing. The paper also demonstrates that their approach outperforms prior works in several functional approximation domains.

**Weaknesses:**

MAJOR:

- Theorem 1 is stated very differently in the Appendix (lines 33-38) compared to the main paper (lines 153-155). In particular, the one in the Appendix adds 2 more assumptions (lines 35-36). This makes the one in the paper severely misleading. This also decreases the strength of this theorem since there almost never is only a single optimal policy in RL (optimal policies are rarely unique), and strictly increasing utility functions further limits the applicability of the proposed framework.

- The paper limits the definition of distributional MORL to non-decreasing utility functions, and the proposed algorithm for finding DPO policies is limited to non-decreasing utilities with objectives bounded by $[0 ~ 1]^K$. While this still covers a wide range of utilities, this severely limits the applicability of the proposed approach.

- No proof is given to show what percentage volume of the Pareto front is covered by the set of optimal policies obtained by the proposed approach. In fact, there doesn't seem to be a reason why we should expect the specific diversity metric given in equation (3) to give us utility functions that lead to DPO policies covering any significant fraction of the Pareto frontier. This is a major missing component of the paper since its operational goal is to extend MORL to DPO policies.

- [1] seems closely related to this paper as it also proposes a distributional view of MORL, and its implementation is not constrained to non-decreasing utilities over objectives bounded by $[0 ~ 1]^K$. However, this paper does not include it in its baselines, nor does it discuss or even cite it.


MINOR

- f(z) is missing on line 7 of the proof of Lemma 1


[1] Abdolmaleki, Abbas, et al. "A distributional view on multi-objective policy optimization." International conference on machine learning. PMLR, 2020.

**Questions:**

It would be great if the authors can address the major weaknesses I outlined above. I am happy to increase my score if they are properly addressed, as I may have misunderstood pieces of paper. In addition to those:

- Why is the set of utility functions restricted to non-decreasing ones? Is this a fundamental limitation of the proposed framework?

- Why is the given Diversity-based Objective Function expected to lead to policies that cover significant volume of the Pareto front? The authors should provide a proof of how well their approach approximates the Pareto front, or at the very least a detailed discussion of it.

- How is the number of utility functions to learn ($M$) determined? Is this just an arbitrary hyperparameter? If so, how does its choice affect how well the approach approximates the Pareto front?

- Why is the given definition of DPO policies to be preferred over that of [1]? Can the authors compare against MO-MPO from [1]?

- In the proof of Lemma 3, the authors say "Suppose π is a Pareto-Optimal policy, then π is the optimal policy under some linear combination of rewards with weight w ≻ 0". Why is this true? Is the lemma maybe assume linear utility functions (if yes this needs to be made explicit)?

**Limitations:**

While there is no limitations section, I believe the authors have adequately addressed the limitations of their work sparsely throughout the paper (besides the ones I mentioned in the weaknesses section). It will be preferable if a limitations section is added to discuss the ones the authors feel are most relevant.

---

> ### Author Rebuttal · Authors · 2023-08-10
>
> To Reviewer GTy6:
> Thanks for your valuable feedback! We provide the response to each of your questions as follows.
>
> **Q1:** “Why is the set of utility functions restricted to non-decreasing ones? Is this a fundamental limitation of the proposed framework?” and “the proposed algorithm for finding DPO policies is limited to non-decreasing utilities with objectives bounded by $[0, 1]^K$ ”
>
> **A1:** This non-decreasing condition is foundational to MORL, aiming to optimize all objectives simultaneously without diminishing any. This basic premise for non-decreasing utility functions is articulated in works like Definition 7 in [2], Definition 3 and 5 in [3], and Section 3.1 in [4]. The normalization of utility function outputs to [0, 1] is a prevalent technique to stabilize model optimization and policy learning across varied objectives, which might span different numerical ranges. This is not a limitation per se but a design choice.
>
> **Q2:** “there doesn't seem to be a reason why we should expect the specific diversity metric given in equation (3) to give us utility functions that lead to DPO policies covering any significant fraction of the Pareto frontier.”
>
> **A2:** At the methodology level, our method (DPMORL) optimizes policies to maximize the expected utility on a diverse set of non-linear utility functions, where Theorem 1 shows that these policies are DPO policies when they have optimal expected utility. At the experimental level, we show in Figure 1 and Figure 2 of the global rebuttal PDF file that DPMORL can learn optimal policies under a diverse set of non-linear utility functions, where the policies cover a diverse set of optimal return distributions. Therefore, we show DPMORL can learn a diverse set of optimal return distributions at both method and experimental levels.
>
> **Q3:** “How is the number of utility functions to learn (M) determined?”
>
> **A3**: In our methodology, the quantity of utility functions aligns with learned policies. This count is a predetermined parameter for MORL scenarios. For fairness in our experiments, we ensured that all baselines method and our method share the same number of learned policies.
>
> **Q4:** “[1] seems closely related to this paper as it also proposes a distributional view of MORL” and “Why is the given definition of DPO policies to be preferred over that of [1]?”
>
> **A4:** Thank you for highlighting the work in [1]. We acknowledge this work's importance in advancing the MORL field by introducing a distributional perspective. However, we'd like to clarify some misunderstandings.
>
> 1.  **Fundamental Conceptual Differences**: Our work introduces the concept of DPO policies to ensure that the policies we derive can accommodate a wide range of distributional preferences in the **return space**, capturing nuances often missed when focusing purely on expected returns. In contrast, [1] proposes a scale-invariant approach to MORL. While this is a valuable perspective, it emphasizes the **action space** and combining objectives in distribution space without necessarily prioritizing the distributional nature of the returns themselves.
> 2. **Implementation Details**: Our approach is grounded in the concept of stochastic dominance for multivariate distributions to learn desirable DPO policies. On the other hand, [1] introduces MO-MPO, characterized as a two-tier policy enhancement procedure. It's pivotal to understand that while MO-MPO is a technique, DPO serves as the objective in our research.
> Given the distinct ambitions and approaches of the two papers, we did not include [1] as one of the baselines in the experiments. Nonetheless, we appreciate the academic merit of comparative discourse. In acknowledgment of the broader academic community's interests, we are glad to enrich our revisions with discussions [1] with our contributions.
>
> **Q5:** “In the proof of Lemma 3, the authors say "Suppose π is a Pareto-Optimal policy, then π is the optimal policy under some linear combination of rewards with weight w ≻ 0". Why is this true? Is the lemma maybe assume linear utility functions (if yes this needs to be made explicit)?”
>
> **A5:** This is correct from Definitions 1 and 4 in [3]. For Lemma 3 “Any Pareto-Optimal policy is a Distributionally Pareto-Optimal policy”, we provide a better proof sketch for the lemma from scratch as follows: if policy π is Pareto-optimal, then its multidimensional expected returns are not dominated by any other policies (see Definition 2.1 in [4]). Then, the return distribution mu(π) of policy π is not dominated by the return distribution mu(π’) of any other policies π’. Therefore, policy π is a Distributionally Pareto-Optimal policy. Lemma 3 has no other assumptions, and does not assume linear utility functions.
>
> **Q6:** “Theorem 1 is stated very differently in the Appendix (lines 33-38) compared to the main paper (lines 153-155). In particular, the one in the Appendix adds 2 more assumptions (lines 35-36). This makes the one in the paper severely misleading.”
>
> **A6:** We appreciate your feedback, where the statement in Theorem 1 lacks the required assumptions, and we will add the assumptions in our paper. Meanwhile, we note that either assumption “π is the only optimal policy” or assumption “utility function f is strictly increasing” can make the claim of Theorem 1 correct, which makes the conclusion adaptable to many more scenarios (than the case where both the assumption has to be satisfied).
>
> ## References
> [1] A distributional view on multi-objective policy optimization. ICML 2020.
>
> [2] A Survey of Multi-Objective Sequential Decision-Making. Journal of Artificial Intelligence Research, 2013.
>
> [3] A practical guide to multi‑objective reinforcement learning and planning. AAMAS, 2022.
>
> [4] Prediction-guided multi-objective reinforcement learning for continuous robot control, ICML 2020.

---

> > ### Author Response · Authors · 2023-08-15
> > **Further Discussions**
> >
> > Dear Reviewer GTy6,
> >
> > We greatly appreciate the time and effort you have invested in reviewing our work. We carefully consider your comments and have addressed each of your concerns and questions in the rebuttal. As the discussion period deadline is approaching, we kindly hope that you can read the response, and consider the new information and experiment results provided. We are open to further discussions and would be happy to provide any additional clarifications if needed.

---

> > > ### Author Response · Authors · 2023-08-21
> > > **Looking forward to your response!**
> > >
> > > Thank you again for your time and effort in reviewing this work! We hope that the weaknesses you outlined were adequately addressed in our rebuttal. Since the author-reviewer discussion will be closed within 8 hours, we hope you can take a moment to review our responses and let us know if there are any outstanding issues that make you consider this paper as borderline reject. Your feedback is crucial in helping us address any remaining concerns and improve the quality of our work.

---

> > > > ### Comment · Reviewer_GTy6 · 2023-08-22
> > > > **Score updated to reflect the current state of the paper given the rebuttal**
> > > >
> > > > Thank you to the authors for the detailed reply and sorry for taking a while to respond. I have read it, the general response, and the rebuttal for the other reviewers. Given all the clarifications provided, I have increased my score to a borderline accept as I believe that best fits the current work.
> > > >
> > > > Here are a few important points I believe would improve the paper (which includes responses I was not convinced by).
> > > >
> > > > - **A1** My point stands. There's a large number of works in MORL that do not make that assumption (see [r1,r2] and their related works). I understand that the survey paper [2] says that the assumption is a minor one, but that's a very old survey paper. That assumption effectively solves the MORL problem by assumption. I suggest the authors cite [2] when making that assumption (to show that there's a part of the community that is fine with it), but also acknowledge and cite works that do not make that assumption.
> > > >
> > > > - **A1** "The normalization of utility function outputs to [0, 1] is a prevalent technique to stabilize model optimization and policy learning across varied objectives, which might span different numerical ranges". Can the authors provide citations that support this claim? For example [r1] does not make that assumption. The normalization of the utility function changes the optimal policy, so you are indeed restricting yourself to a smaller set of Pareto optimal policies.
> > > >
> > > > - **A6** Thanks for agreeing to make that correction, as that is very important. The continuation of the authors answer here didn't help. “π is the only optimal policy” is not true in almost all MDPs and is a trivial way to make Theorem 1 true. Hence the real assumption here is that “utility function f is strictly increasing”.
> > > >
> > > > - **A3** The number of utility functions to learn $M$  being a hyperparameter underscores my main issue with this work.
> > > >   - By assumption you only consider strictly increasing utility functions.
> > > >   - By assumption you only consider utility functions in the range [0 1].
> > > >   - By assumption you only learn $M$ such utility functions where $M$ is given.
> > > >   - I believe these trivialise the MORL problem.
> > > >   - **I suggest the authors do one of the following in future versions of this work:**
> > > >     - For a given budget $M$, theoretically show that the specific diversity metric given in equation (3) gives us utility functions that have the best coverage of the Pareto frontier.
> > > >     - Provide a method for determining $M$ such that the resulting optimal policies cover a higher fraction of the Pareto frontier compared to prior works like [r1].
> > > >
> > > > All that being said, as reflected by the fact that I increased my score, I really like the proposed approach and believe the strengths of this paper outweigh its flaws.
> > > >
> > > > [r1] Alegre, Lucas Nunes, Ana Bazzan, and Bruno C. Da Silva. "Optimistic linear support and successor features as a basis for optimal policy transfer." International Conference on Machine Learning. PMLR, 2022.
> > > >
> > > > [r2] Abdolmaleki, Abbas, et al. "A distributional view on multi-objective policy optimization." International conference on machine learning. PMLR, 2020.

---

### Official Review · Reviewer_JBHP · 2023-07-09

**Soundness:** 3 good
**Presentation:** 3 good
**Contribution:** 2 fair
**Rating:** 5
**Confidence:** 4

**Summary:**

This work introduces distributional Pareto-optimality for multi-objective reinforcement learning. Multi-objective RL (MORL) is typically formulated as to find Pareto optimality of all objectives.

To build the ground, the paper defines stochastic dominance for multivariate distribution and then stochastic dominance for policies. Distributional Pareto-optimality can then be formulated.

The authors propose a practical algorithm for addressing the distributional MORL problem. First, a non-decreasing NN is used (as the Q network in MARL paper QMIX) as the utility function. Then a new objective function can be defined and creates a reward function for RL. Therefore, we can iteratively train the utility function and policies.

Experiment in multi-objective gymnasium shows the effectiveness of the method.

**Strengths:**

1. The paper addresses the distributional preferences in multi-objective RL, which is an understudied problem.
2. The paper formulates a framework and propose practical method to address the problem.

**Weaknesses:**

 1. The paper is self-contained and there is no particular worth-noting weaknesses. It would be great to see if the proposed method can be applied to other more realistic environments, such as safety-aware autonomous driving. I believe driving is a decision-making task that has many objectives that should be considered.

**Questions:**

1. The authors list safe RL as a special case to MORL. That would great if we can apply the proposed method and other baselines to safe RL benchmark such as Safety Gym to see the performance and also compared to other safe RL baselines. This is because safe RL is a domain with huge social impact and thus working on those tasks can improve the impact of the proposed method.
2. In the DiverseGoal environment, how we get the distribution of reward? Is it some kinds of Normal distribution predefined?

**Limitations:**

The authors do not address the limitation at all.

---

> ### Author Rebuttal · Authors · 2023-08-10
>
> To Reviewer JBHP:
> Thanks for your valuable feedback! We provide the response to each of your questions as follows.
>
> **Q1:** “The authors list safe RL as a special case to MORL. That would great if we can apply the proposed method and other baselines to safe RL benchmark such as Safety Gym to see the performance and also compared to other safe RL baselines.”
>
> **A1:** In Figures 1 and 2 of the global rebuttal PDF file, we show experimental results demonstrating that DPMORL can effectively optimize policies with utility functions that characterize safety properties. Specifically, the DeepSeaTreasure environment in Figure 2 resonates with safe RL dynamics, balancing between time penalties and task rewards. With our method, users can provide different utility functions derived from safety constraints, and optimize the expected utility via DPMORL for obtaining policies that satisfy the constraint. Due to the one-page limitation of the rebuttal pdf, we cannot provide the comparison results on more environments with safe RL baselines, but we're committed to delving deeper into this in our revised manuscript.
>
> **Q2:** “In the DiverseGoal environment, how we get the distribution of reward? Is it some kinds of Normal distribution predefined?”
>
> **A2:** Yes, in the DiverseGoal environment, the reward distribution associated with each goal position is a pre-defined Normal distribution. Contrasted with standard environments where the reward functions are deterministic, in our DiverseGoal environment, the reward from distinct objectives is sampled from unique Normal distributions. Consequently, the approach necessitates acquiring a policy set embodying the distributional attributes of returns to cater to diverse user preferences. For more environments in MO-Gymnasium, we provide the same case study results on the effectiveness of our method for optimizing a diverse set of non-linear utility functions under stochastic rewards in Figures 1 and 2 of the global rebuttal PDF file.

---

> > ### Comment · Reviewer_JBHP · 2023-08-15
> >
> > Thanks for the response.

---

> > > ### Author Response · Authors · 2023-08-21
> > > **Thank you for the comments!**
> > >
> > > Thank you for your constructive comments and positive feedback! We are delighted that our rebuttal responses addressed your questions and concerns. Since the author-reviewer discussion will be closed within 8 hours, we are more than happy to provide any additional information or explanations if there are any other concerns that make you still consider this paper as borderline accept. Your feedback is crucial in helping us address any remaining concerns and improve the quality of our work.

---

### Official Review · Reviewer_y77J · 2023-07-25

**Soundness:** 3 good
**Presentation:** 3 good
**Contribution:** 2 fair
**Rating:** 5
**Confidence:** 4

**Summary:**

In Multi-Objective Reinforcement Learning (MORL), the goal is to learn Pareto optimal policies that achieve a balance among multiple conflicting objectives while considering the user's preferences. Existing MORL methods optimize the Pareto frontier by using a single value obtained from the linear combination of a multi-dimensional reward vector representing the user's preferences. In this paper, instead of the traditional approach, the authors interpret the problem from a return distribution perspective and consider the uncertainty in returns to better capture the user's preference distribution. Furthermore, they propose an algorithm that generates various utility functions, enabling the learning of Pareto optimal policies for diverse preferences.

**Strengths:**

- Instead of considering the traditional approach of single expected value in MORL, the paper proposes a Distributional Pareto-Optimal (DPO) method by defining the return as a distribution. This approach enables capturing complex preferences and allows for a more nuanced representation of utility functions.
- The paper defines Stochastic Dominance (SD) and uses it to define Pareto-Optimal policies. Furthermore, it provides a mathematical proof that the pareto-optimal policy is Distributional Pareto-Optimal policy, which means that their return distributions of each policy cannot dominate that of another.
- The author presents DPMORL algorithm that iteratively generates various non-linear, non-decreasing utility functions in cases where Stochastic Dominance does not exist and then train DPO policies with the generated utility functions. From these utility functions, proposed algorithm identifies Pareto-Optimal policies. By generating various utility functions, the algorithm can achieve to get more flexible and diverse policies set that can accommodate a wide range of preferences.

**Weaknesses:**

- The main experiment shows superior performance of the proposed algorithm compared to existing MORL methods but seems to do not fully explain the motivation(ability to capture users' complex preferences and intentions in a more detailed manner). Including additional experiments or toy examples such as diversity of utility function heatmaps between the return distribution approach and the conventional weighted sum method would substantiate the authors' motivation.
- The proposed loss function aims to generate various utility functions, and it consists of three components: equations (1), (2), and (3). Providing explanations for each of these components would make it more readable, describing their purposes and how they contribute to creating diverse utility functions.
- It would be beneficial to include an ablation study. For instance, in the main experiment, training 20 utility functions and policies may show differences in evaluation values with changes in the size of N. By conducting experiments with various N sizes and adding the results to the study, it could provide further support for the author's insistence.
- Additionally, the paper proposes a method to increase the diversity of utility functions by iteratively generating them. In Appendix C Table 1, they applied N^{iter}=2 iterations to create a broader range of utility functions and trained policies based on them. However, the experimental results show that the performance does not exhibit significant differences compared to the experiments where utility functions were generated only once. Increasing N^{iter} is expected to generate a more diverse set of utility functions, enabling the learning of various Pareto-optimal policies. Consequently, this is likely to further enhance the "Constraints Satisfaction" score.
- Minor typos: (1) Line 192 “We maximize” → “We minimize”,  (2) Appendix line 99 “standard derivation” → “standard deviation”


**Questions:**

- Can we design utility functions in a real-world setting that consider individual preferences? For instance, in the case of autonomous driving, can we provide information that highlights safety to encourage the creation of utility functions that prioritize safety significantly?
- I wonder how the algorithm performs even in cases where the multiple objectives in the Continuous Control environment experiments are conflicting. For instance, in the case of HalfCheetah, which has two multi-objectives, "run" and "jump," in the experiments, I wonder if the algorithm can still learn Pareto-optimal policies based on preferences when an additional conflicting objective, such as "run backward," is introduced.
- The authors conducted MORL experiments on various environments in the main experiments. It would be better if they provide more detailed explanation with some examples of what objectives in each environment and how many objectives are there. 2D graphs of Figure 5 and Appendix Figure 1 seem that the environments have two objectives each. I wonder about some results of the experiments with three or more objectives.
- Appendix Figure 1 depicts the return distribution of the set of policies learned by DPMORL on each environment, and Return 1 and 2 seem to mean returns for two objectives. It would be helpful if authors clarify explanation of what each return means, and what preference each policy has.


**Limitations:**

Author provides the limitations of their works appropriately, and I think there is no potential negative societal impact.

---

> ### Author Rebuttal · Authors · 2023-08-10
>
> To Reviewer y77J:
> Thanks for your valuable feedback! We provide the response to each of your questions as follows.
>
> **Q1:** “The main experiment shows superior performance of the proposed algorithm compared to existing MORL methods but seems to do not fully explain the motivation”
>
> **A1:** The motivation of the paper is to optimize for a diverse set of policies for optimal return distributions, which is achieved by optimizing policies on a diverse set of non-linear utility functions. In the main paper, we provide case studies in Figure 4 in the DiverseGoal environment, which shows that DPMORL can obtain policies with a diverse set of return distributions with different utility functions to optimize, supporting the motivation on a toy example. Here, we will show the additional experimental results on harder environments in MO-Gymnasium for the same case study, which is shown in Figures 1 and 2 of the global rebuttal PDF file. In the ReacherBullet and DeepSeaTreasure environment, when providing different generated utility functions for the policies, DPMORL is able to optimize the policy to obtain policies with optimal non-linear utility, covering a diverse set of optimal return distributions, which directly supports our motivations. We will add the additional case studies in Figure 1 and Figure 2 to our paper to better support our motivation in the revision.
>
> **Q2:** “equations (1), (2), and (3). Providing explanations for each of these components would make it more readable, describing their purposes and how they contribute to creating diverse utility functions.”
>
> **A2:**  Briefly, equation 1 quantifies the output difference between two utility functions, and accordingly measures the smallest distance from one utility function to any other utility function; equation 2 quantifies the average slope difference between two utility functions, and accordingly measures the smallest distance from one utility function to any other utility function, and equation 3 is a weighted combination of equation 1 and 2, which is optimized to maximize the dissimilarity of the current utility function. We will add the explanation to our paper in the revision.
>
> **Q3:** “conducting experiments with various N sizes and adding the results to the study”
>
> **A3:** We show the experimental results of various N sizes in the Table 2 of the global rebuttal PDF file. As N becomes larger, DPMORL has better Expected Utility, and Hypervolume.
>
> **Q4:** “Increasing N^{iter} is expected to generate a more diverse set of utility functions”
>
> **A4:** In Table 1 in the Appendix, DPMORL in the second iteration has improved performance in 11 of the metrics compared to the first iteration, while having 7 metrics in worse performance. In ReacherBullet, we also find DPMORL in the second iteration having an overall better performance than in the first iteration. We will add the additional experimental results in the revised version of our paper.
>
> **Q5:** “can we provide information that highlights safety to encourage the creation of utility functions that prioritize safety significantly?”
>
> **A5:** In Figures 1 and 2 of the global rebuttal PDF file, we show experimental results demonstrating that DPMORL can effectively optimize policies with utility functions that characterize safety properties. The DeepSeaTreasure environment of Figure 2 aligns well with safe RL settings, where the two returns are time penalty and task reward respectively. With our method, users can provide different utility functions derived from safety constraints, and optimize the expected utility via DPMORL for obtaining policies that satisfy the constraint. Due to the one-page limitation of the rebuttal pdf, we did not provide the comparison results on more environments with safe RL baselines, and we will add more results on this topic in the revision.
>
> **Q6:** “I wonder about some results of the experiments with three or more objectives.” and “in the case of HalfCheetah, which has two multi-objectives, ‘run’ and ‘jump,’ in the experiments, I wonder if the algorithm can still learn Pareto-optimal policies based on
> preferences when an additional conflicting objective, such as ‘run backward,’ is introduced.”
>
> **A6:** The DPMORL algorithm can be applied well for three or more objectives. We have added experiments with 3 objectives under Hopper and FruitTree environment, as illustrated in the Figure 3 and Table 1 of the global rebuttal PDF file. Figure 3 demonstrates that DPMORL learns better 3-dimensional return distributions than GPI-PD, a competitive MORL baseline method; Table 1 demonstrates that DPMORL achieves better performance in different MORL metrics compared to baseline methods under 3 dimensional reward functions. Regarding the HalfCheetah scenario, introducing a diametrically opposing objective, like “running backward”, as an antithesis to “running forward” would render any arbitrary policy as a Pareto-optimal policy. This diverges from the optimization goal of MORL. Hence, we selected widely accepted and pragmatic MORL environments and targets for our experiments.
>
> **Q7:** “Appendix Figure 1 depicts the return distribution of the set of policies learned by DPMORL on each environment, and Return 1 and 2 seem to mean returns for two objectives. It would be helpful if authors clarify explanation of what each return means, and what preference each policy has”
>
> **A7:** We add the brief explanations to the meaning of reward here: for DeepSeaTreasure, MountainCar, and HalfCheetah, the two objectives are respectively task reward (gathering treasure, climb mountain and running) and energy consumption; for FruitTree, Hopper, and Reacher, the two objectives are the reward of reaching two different types of positions. We will add the clarifications in the appendix of our paper. Our case studies (Figure 2) in the global rebuttal PDF file show the preference for a part of learned policies, and we will add preferences for more policies in our revised manuscript.

---

> > ### Author Response · Authors · 2023-08-15
> > **Further Discussions**
> >
> > Dear Reviewer y77J,
> >
> > We greatly appreciate the time and effort you have invested in reviewing our work. We carefully consider your comments and have addressed each of your concerns and questions in the rebuttal. As the discussion period deadline is approaching, we kindly hope that you can read the response, and consider the new information and experiment results provided. We are open to further discussions and would be happy to provide any additional clarifications if needed.

---

> > > ### Comment · Reviewer_y77J · 2023-08-21
> > >
> > > I read all your comments including other reviewers. I really appreciate your reply about my questions. The paper will be more competitive and solid when you add supplementary experiments mentioned above.

---

> > > > ### Author Response · Authors · 2023-08-21
> > > > **Thank you for the comments!**
> > > >
> > > > Thank you for your constructive comments and positive feedback! We are delighted that our rebuttal responses addressed your questions and concerns. Since the author-reviewer discussion will be closed within 8 hours, we are more than happy to provide any additional information or explanations if there are any other concerns that make you still consider this paper as borderline accept. Your feedback is crucial in helping us address any remaining concerns and improve the quality of our work.

---

### Author Rebuttal · Authors · 2023-08-10

We express our sincere gratitude to all reviewers for their constructive feedback and insightful suggestions. In this global response, we include a supplementary PDF comprising additional experimental outcomes. We summarize the experiment result as follows:

### **Figure 1: Enhanced Case Studies for DPMORL on ReacherBullet Environment**

In the case study, we show that DPMORL can effectively learn optimal policies under a diverse set of non-linear utility functions under the complex control environment (ReacherBullet), and the policies cover a wide range of optimal return distributions. We randomly generate 7 different non-linear utility functions with our method, and learn 7 policies by DPMORL to maximize expected utility for each utility function. We use the first two reward functions (two goal-reaching rewards) in MO-Gymnasium.

The results show that our algorithm effectively finds policies that maximize the expected utility for different non-linear utility functions, and the learned policies cover broader areas on the return distributions compared to the baseline methods.

### **Figure 2: Enhanced Case Studies for DPMORL on DeepSeaTreasure Environment**

In the case study, we show that DPMORL can effectively learn optimal policies under a diverse set of non-linear utility functions under the DeepSeaTreasure environment. We randomly generate 7 different non-linear utility functions with our method, and learn 7 policies by DPMORL to maximize expected utility for each utility function. We use the same two-dimensional reward functions (two goal-reaching rewards) as in MO-Gymnasium.

The results show that our algorithm effectively finds policies that maximize the expected utility for different non-linear utility functions.

The DeepSeaTreasure environment of Figure 2 aligns well with safe RL settings, where the two returns are time penalty and task reward respectively. With our method, users can provide different utility functions derived from safety constraints, and optimize the expected utility via DPMORL for obtaining policies that satisfy the constraint.

### **Figure 3: Return distribution of DPMORL on more-than-two Dimensional Objectives**

In this experiment, we show the 3-dimensional return distribution of the set of policies learned by DPMORL under Hopper and FruitTree environments. The results demonstrate that DPMORL can effectively learn DPO policies under more than two-dimensional reward functions. Also, DPMORL learns better 3-dimensional return distributions than GPI-PD, a competitive MORL baseline method.

### **Table 1: Comparative Analysis for DPMORL on more-than-two Dimensional Objectives**

In this experiment, we compare the performance of our method on more than two dimensions of reward with baseline methods on four MORL evaluation metrics. We run our experiment on Hopper and FruitTree environments with 3-dimensional reward function, following the reward settings in MO-Gymnasium. We measure four MORL metrics: conditional value at risk, constraint satisfaction and variance Objective, and expected utility. These metrics are detailed in Appendix C. DPMORL achieves the best performance compared to all baseline methods under 3d reward functions.

### **Table 2: Ablation Analysis with Variable Policy Counts**

In this experiment, we show the ablation studies of our method with a varying amount of learned policies ($N$). We can see that as the number of policies increases, both expected utility and hypervolume increase, which means DPMORL can indeed obtain better performance as the number of policies increases. On the other hand, DPMORL has obtained promising results when there exists only 5 policies, which verify the effectiveness of DPMORL among all environments.

---

### Decision · Program_Chairs · 2023-09-21

**Decision:**

Accept (poster)

**Comment:**

This paper considers the problem of learning Pareto-optimal policies under risk-sensitive risk criteria. A method for generating diverse monotonic utility functions is proposed, and distinct policies are then trained on these reward functions, as a way of approximating policies at the Pareto frontier with respect to the risk-sensitive criteria considered.

All reviewers recommend acceptance for the paper. There is a general consensus that the authors are tackling an interesting problem, that there is merit in the approach they propose, and that the paper is well written.

Several important concerns were also noted during the review process, including:
 - Reviewer GTy6 notes that the authors introduced additional assumptions into the statement of Theorem 1 in the course of proving it in the appendix. It is crucial that the main paper statement is amended to reflect these assumptions, and some discussion is given as to the strength of the assumptions/applicability of the theorem in light of them. In particular, as Reviewer GTy6 notes, a unique optimal policy is a strong assumption.
 - From a narrative point of view, several reviewers felt that the paper could do more to connect the proposed approach to the stated motivations relating to capturing complex preferences.
 - The focus of the main paper experiments is primarily on qualitative aspects (i.e. visualization of learned utility functions) and performance (Tables 1 and 2). Several reviewers noted that the experimental work in the paper could be significantly strengthened with ablative experiments. I strongly agree with this, and think this is the clearest path to improving the paper. In particular, experiments that aim to develop an understanding of the hyperparameters of the method, as well as situations in which the method may not work as well (such as the case of conflicting objectives, as mentioned by Reviewer y77J) would strengthen the paper considerably.
 - Reviewer GTy6 also notes that there is relatively little theoretical/empirical investigation into the diversity objective in Equation (3), and why this (rather than some other objective) should be expected to provide a wide covering of Pareto-optimal policies.

The authors have posted a detailed rebuttal in response to these questions, including additional experiments in the rebuttal pdf. After discussion, the reviewers believe the strengths of the paper outweigh the concerns described above.

In my own view, there are a few other aspects I would have liked more commentary on, which should be straightforward to address for the authors:
 - The specific types of risk-sensitivity the authors consider are expectations of non-linear functions of the return. I believe this excludes common risk measures such as CVaR, and I would have liked to see this limitation discussed.
 - It would be good to make Definition 1 more precise by being clear about exactly what is meant by non-decreasing f in the multivariate case.
 - Reference [19] for distributional MPO is not correct. A more appropriate reference may be Acme (Hoffman et al., 2020).

The recommendation overall is to accept the paper. I strongly encourage the authors to take on board the constructive feedback provided by the reviewers in the course of preparing the camera-ready version of the paper.